Analysis

# Ecological dynamics of Enterobacteriaceae in the human gut microbiome across global populations

Qi Yin [1,2], Ana C. da Silva [1], Francisco Zorrilla [3], Ana S. Almeida[4,5], Kiran R. Patil[3] & Alexandre Almeida [1]✉

Gut bacteria from the Enterobacteriaceae family are a major cause of opportunistic infections worldwide. Given their prevalence among healthy human gut microbiomes, interspecies interactions may play a role in modulating infection resistance. Here we uncover global ecological patterns linked to Enterobacteriaceae colonization and abundance by leveraging a large-scale dataset of 12,238 public human gut metagenomes spanning 45 countries. Machine learning analyses identified a robust gut microbiome signature associated with Enterobacteriaceae colonization status, consistent across health states and geographic locations. We classified 172 gut microbial species as co-colonizers and 135 as co-excluders, revealing a genus-wide signal of colonization resistance within *Faecalibacterium* and strain-specific co-colonization patterns of the underexplored *Faecalimonas phoceensis*. Co-exclusion is linked to functions involved in short-chain fatty acid production, iron metabolism and quorum sensing, while co-colonization is linked to greater functional diversity and metabolic resemblance to Enterobacteriaceae. Our work underscores the critical role of the intestinal environment in the colonization success of gut-associated opportunistic pathogens with implications for developing non-antibiotic therapeutic strategies.

The human gut microbiota comprises a diverse microbial community playing a fundamental role in human health. An increasing number of studies are revealing the importance of a healthy gut microbiome not only for digestion and immune regulation, but also in controlling the colonization of exogenous and opportunistic pathogens[1,2]. Consequently, disruption to the human gut microbiome composition and function has been associated with numerous pathologies[3-5].

Although the intestinal tract is colonized by a varied community of commensal microorganisms with beneficial roles to human health, many gut microbial species also have the potential to cause disease. Species from the Enterobacteriaceae family, such as *Escherichia coli* and *Klebsiella pneumoniae*, represent opportunistic pathogens with the potential to cause severe, life-threatening infections. An overabundance of Enterobacteriaceae in the gut not only increases infection risk but has also been linked to non-communicable diseases such as Crohn's disease[6] and even a higher all-cause mortality[7]. Notably, multidrug- and extended-spectrum β-lactamase (ESBL)-producing Enterobacteriaceae have been classified as priority 1 pathogens by the World Health Organization. Moreover, transmission rates of ESBL-producing Enterobacteriaceae in household settings (23% for *E. coli* and 25% for *K. pneumoniae*) were found to surpass those in healthcare environments[8]. Therefore, there is a global need to control the outgrowth and

[1]Department of Veterinary Medicine, University of Cambridge, Cambridge, UK. [2]College of Public Health, Chongqing Medical University, Chongqing, China. [3]Medical Research Council Toxicology Unit, University of Cambridge, Cambridge, UK. [4]GIMM - Gulbenkian Institute for Molecular Medicine, Lisbon, Portugal. [5]Faculdade de Medicina, Universidade de Lisboa, Lisbon, Portugal. ✉e-mail: aa2369@cam.ac.uk

transmission of Enterobacteriaceae in the human population beyond traditional antibiotic-based therapies. Microbiome-derived therapeutics utilizing beneficial species and/or functions found in the human gut represent promising alternative strategies to mitigate pathogen colonization. The most successful example of a microbiome-based therapy thus far has been in the treatment of *Clostridioides difficile* infection, where faecal microbiota transplantation from healthy donors to infected patients has been shown to resolve ~90% of cases[9].

Most research studying the pathogenic potential, antimicrobial resistance and evolution of Enterobacteriaceae has focused on clinical isolates, limiting our understanding of their ecology within the surrounding intestinal microbiome. Exploring the relationship between Enterobacteriaceae and the gut microbiome may provide ecological insights into controlling the colonization of Enterobacteriaceae and ultimately reducing disease risk. Advancements in sequencing technologies and culture-independent metagenomic analyses have provided substantial improvements in our ability to characterize the composition and diversity of the human microbiome. However, previous studies investigating the relationship between Enterobacteriaceae and the gut microbiome have been limited by small sample sizes and/or have been solely based on 16S rRNA genotyping[10,11], a technique with reduced taxonomic and functional resolution. Furthermore, recent metagenomic developments have enabled the creation of a more comprehensive sequence catalogue of the human gut microbiome, comprising >200,000 genomes that include thousands of previously uncharacterized species[12]. This has now opened opportunities to uncover the intricate dynamics and interactions of opportunistic pathogens within the human gut microbiome at an unprecedented resolution.

Here we perform a large-scale, high-resolution metagenomic analysis investigating the ecology of Enterobacteriaceae within the human gut microbiome across >12,000 human gut metagenomic samples distributed worldwide. We identified over 300 candidate species significantly associated with Enterobacteriaceae colonization dynamics and used gene functional analyses combined with metabolic modelling to obtain mechanistic insights into these interspecies interactions. Our results expand our understanding of the role of the microbiome and the intestinal environment in the colonization success and abundance of Enterobacteriaceae in the human gut.

## Results

### Distribution of Enterobacteriaceae worldwide
To perform a comprehensive global characterization of human gut microbiome signatures linked to Enterobacteriaceae colonization, we retrieved 12,238 public human gut metagenomic samples from 65 studies across 45 countries (Fig. 1 and Supplementary Table 1). Samples were collected primarily from Europe (*n* = 4,284, 35%) and North America (*n* = 3,367, 27.5%), followed by Asia (*n* = 2,844, 23.2%) and Africa (*n* = 1,024, 8.4%). The majority of metagenomic datasets were from adults (*n* = 8,275, 67.6%) and healthy individuals (*n* = 7,606, 62.2%) (Extended Data Fig. 1a).

Using the Unified Human Gastrointestinal Genome (UHGG) catalogue[12], we employed a mapping-based approach to accurately detect the presence and abundance of 4,612 gut microbial species (113 from the Enterobacteriaceae family) in the 12,238 global metagenomes on the basis of their level of breadth, depth and expected coverage (see Methods); the chosen thresholds were further validated with an experimental mock community (Extended Data Fig. 1b). Applying these parameters to synthetic metagenomes showed the detection limit of our metagenomic approach to be within a relative abundance of 0.003–0.01% (Extended Data Fig. 1c). The overall prevalence observed for Enterobacteriaceae was 66%, which is in line with a previous culturing-based surveillance study of *Escherichia coli* found in stool[13]. The distribution of Enterobacteriaceae was generally well balanced across metadata categories (Fig. 1c) and studies (Extended Data Fig. 1d), with a median prevalence of 57% across continents, 67%

across age groups and 71% across different health states. The genera *Escherichia*, *Klebsiella* and *Enterobacter* were the most prevalent across various age groups, health states and continents, with *E. coli*, *Klebsiella pneumoniae* and *Enterobacter hormaechei* representing the most frequent species in their respective genera. The prevalence of *E. coli* was highest among African samples (88%), infant metagenomes (74%) and rheumatoid arthritis patients (96%) (Fig. 2a).

We further investigated the co-distribution of Enterobacteriaceae species across this gut metagenomic collection to characterize patterns of polymicrobial colonization (Fig. 2b). Although most samples were found to be uniquely colonized by *E. coli*, we identified a statistically significant co-colonization of *E. coli* with *K. pneumoniae* predominantly among samples from Asia (observed vs expected proportion of 16.2% vs 10.9%, binomial exact test, *P* < 0.001). Moreover, we detected a significant co-occurrence of *E. coli*, *K. pneumoniae* and *Enterobacter hormaechei* primarily in samples from Africa and Oceania (observed vs expected proportion of 5.3% vs 0.7%, binomial exact test, *P* < 0.001). Overall, these distinct geographic patterns of Enterobacteriaceae co-colonization might be reflective of variations in environmental conditions, dietary habits, lifestyle and/or healthcare practices.

As *E. coli* is the most prevalent Enterobacteriaceae in the gut, we performed a dedicated analysis of the strain diversity of this species. To reduce the effect of host state and better understand the subspecies diversity circulating asymptomatically in the human population, we focused on 5,128 metagenomic samples collected from healthy adults. Through a metagenomic multilocus sequence typing (MLST)[14] we identified 585 distinct *E. coli* sequence types (STs) (Extended Data Fig. 2a). The most prevalent known STs were classified as ST10, ST95, ST131 and ST73, which represent dominant *E. coli* lineages worldwide[13,15,16]. However, 76.5% of detected strains belonged to unknown STs, including two STs (here labelled as 100024 and 100083) found to be among the 10 most frequent STs (Extended Data Fig. 2a). These unknown lineages were found to be overrepresented particularly among samples from Africa (Extended Data Fig. 2b). Given current reference biases towards *E. coli* clinical isolates and the extent of unknown subspecies diversity uncovered here, these results suggest that a substantial global diversity of *E. coli* remains uncharacterized.

### Microbiome structure linked to colonization dynamics
Having access to a global collection of >12,000 human gut metagenomes enabled us to explore the relationship between the gut microbiome composition and Enterobacteriaceae colonization status. First, we built machine learning classifiers to distinguish samples with or without Enterobacteriaceae on the basis of the abundance and prevalence of the remaining non-Enterobacteriaceae microbiome species (Fig. 3a and Extended Data Fig. 3a). We tested three supervised learning methods (ridge regression, random forest and gradient boosting) using as outcome variables the colonization status of Enterobacteriaceae as a whole, or that of *E. coli* and *K. pneumoniae* in particular. Results across all methods and variables showed a consistently good performance (median area under the receiver operating curve, AUROC = 0.788), with gradient boosting outperforming ridge regression and random forest for the classification of Enterobacteriaceae (median AUROC = 0.812), *E. coli* (median AUROC = 0.797) and *K. pneumoniae* status (median AUROC = 0.773). Model performance was consistent between the entire metagenomic dataset and when only considering samples from healthy adults (Extended Data Fig. 3a). Given the variation in gut microbiome composition between geographic regions, we also investigated whether models were generalizable within and across different continents. Focusing on samples from healthy adults, models tested on a per continent basis had overall good performance (AUROC > 0.7 for all continents tested, Extended Data Fig. 3b), with the highest performance observed for samples from Africa. Cross-validation between continents also revealed that models specifically trained on metagenomes from Asia, Europe or North America performed well across other regions

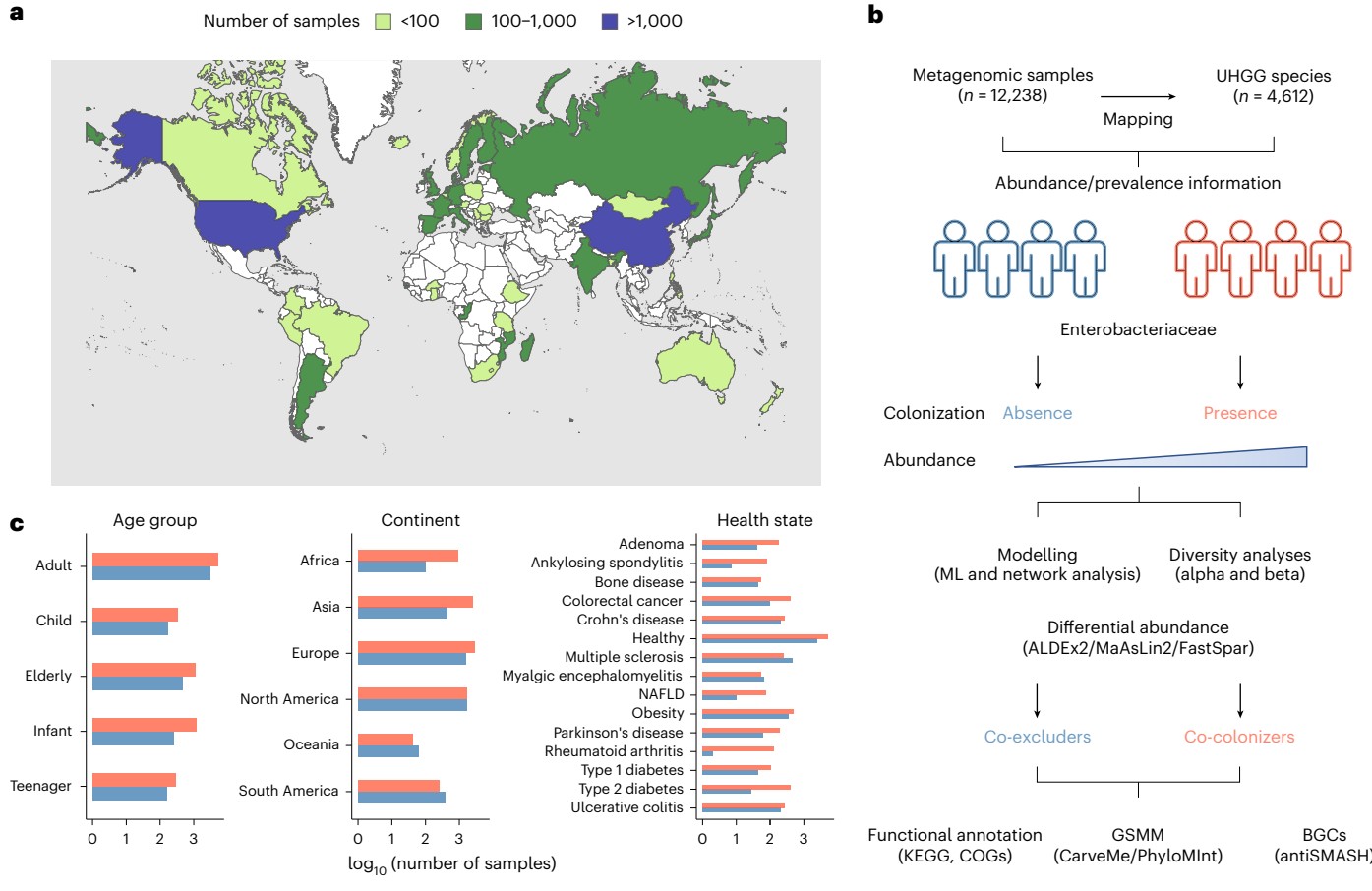

**Fig. 1 | Exploring the global ecological landscape of Enterobacteriaceae.**
**a**, Geographic distribution of the 12,238 human gut metagenomic samples used in this study. **b**, Workflow developed to identify and functionally characterize Enterobacteriaceae co-excluders and co-colonizers. ML, machine learning;

GSMM, genome-scale metabolic modelling; BGCs, biosynthetic gene clusters.
**c**, Metadata distribution of the number of samples where no Enterobacteriaceae species was detected (Absent) or at least one species was detected (Presence).

(AUROC > 0.7 in at least 2 other continents, Extended Data Fig. 3c), showing that models trained on larger sample sizes are more generalizable. Overall, these results indicate that the human gut microbiome harbours compositional differences linked to Enterobacteriaceae colonization, even across different health states and geographic locations.

We next performed diversity analyses to investigate overall community differences relating to Enterobacteriaceae colonization and abundance (Extended Data Fig. 4). Beta diversity estimates revealed higher pairwise distance among samples with Enterobacteriaceae compared with those without (Wilcoxon rank-sum test, $P < 0.0001$, Extended Data Fig. 4a). These differences were independent of alpha diversity and sample read depth, as we observed a low correlation between Enterobacteriaceae abundance and Shannon diversity estimates (Pearson's coefficient of determination, $R^2 = 0.03$, Extended Data Fig. 4b), even after subsampling to 500,000 mapped reads—the minimum depth needed for accurate beta and alpha diversity estimates[17].

To identify the microbiome species associated with Enterobacteriaceae presence (co-colonizers) or absence (co-excluders), we performed a differential abundance analysis on the basis of the intersection of a generalized (ALDEx2 (ref. 18)) and mixed-effects model (MaAsLin2 (ref. 19)), while accounting for study, age group, health state, continent and read depth (see Methods for further details). In addition, beyond investigating the differential abundance of species according to Enterobacteriaceae colonization status at a binary level (presence/absence), we used a network-based approach[20,21] to model Enterobacteriaceae–microbiome co-abundance patterns. We analysed all 12,238

human gut metagenomes, as well as the subset of 5,128 samples from healthy adults to further control for microbiome differences related to age and health state. This revealed 307 prokaryotic species (12% of prevalence-filtered species) significantly associated with Enterobacteriaceae, *E. coli* and/or *K. pneumoniae* colonization and abundance (Fig. 3b, Extended Data Fig. 5a and Supplementary Table 2): 172 were identified as Enterobacteriaceae co-colonizers and 135 as Enterobacteriaceae co-excluders.

At a taxonomic level, species from the orders Lachnospirales, Oscillospirales and Bacteroidales were overrepresented among the co-excluders (Fisher's exact test, adjusted $P < 0.05$). In contrast, co-colonizers were significantly associated with the orders Lactobacillales, Veillonellales and Actinomycetales. Analysis of the 1,000 most prevalent species revealed that 17 bacterial orders contained neither co-excluders nor co-colonizers (Extended Data Fig. 5b), even though two taxa in particular (RF39 and Burkholderiales) were represented by >10 species each. At a species level, 89% of the 307 candidate species showed a consistent signal across datasets (all or healthy adults only) and taxa (Enterobacteriaceae, or *E. coli* and *K. pneumoniae* individually) (Extended Data Fig. 5c), showing that the identified microbiome signatures are robust to differences in host state and analysis resolution. Species from the *Faecalibacterium* genus were among the strongest co-excluders (Fig. 3c and Extended Data Fig. 5d), with one uncharacterized *Faecalibacterium* species (*Faecalibacterium* sp.900539885) identified as the top antagonistic candidate. Previous studies have shown that species from the *Faecalibacterium* genus (for example,

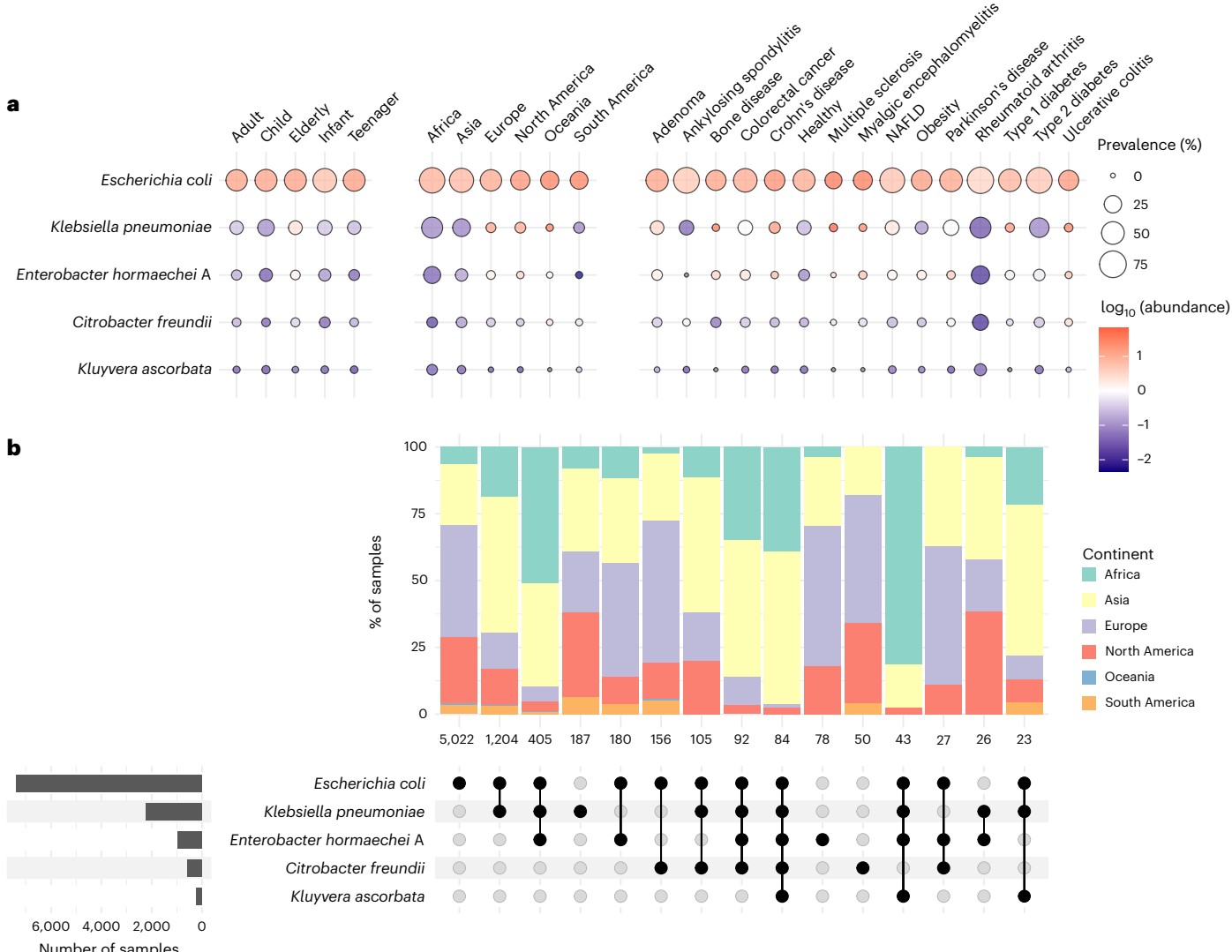

**Fig. 2 | Distribution and diversity of the most prevalent Enterobacteriaceae species. a**, Prevalence and median abundance of representative species from the five most prevalent Enterobacteriaceae genera across different age groups, continents and health states. **b**, Upset plot showing Enterobacteriaceae co-colonization patterns. Vertical bars represent the proportion of samples by continent harbouring the species highlighted in the lower panel. Numbers below the bars indicate sample size. Horizontal bars in the lower left panel show the total number of samples in which each species was detected.

*F. prausnitzii*) carry important beneficial functions in the intestinal tract such as the production of short-chain fatty acids (SCFAs)[22]. This in turn has been shown to directly inhibit the growth of Enterobacteriaceae species, including *E. coli* and *K. pneumoniae*[23]. Thus, the over-representation of *Faecalibacterium* species in gut microbiomes with low levels of Enterobacteriaceae might be reflective of a genus-wide mechanism of colonization resistance. Focusing on the co-colonization patterns, we found that members of the *Intestinibacter*, *Veillonella* and *Enterococcus* genera were the strongest candidates. *E. faecalis* has been previously shown to promote the growth and survival of *E. coli* in vitro and in vivo through the production of L-ornithine[24]. However, the relationship between species of the *Intestinibacter* and *Veillonella* genera with Enterobacteriaceae has not been previously explored and may potentially underlie uncharacterized mechanisms associated with Enterobacteriaceae colonization success and outgrowth.

To further evaluate the clinical relevance of our findings, we compared our results with an independent study that investigated the longitudinal dynamics of carbapenemase-producing Enterobacteriaceae (CPE)[10]. This study tracked for up to 12 timepoints a cohort of CPE-positive individuals that were later decolonized, as well as

CPE-negative household controls (n = 46 participants; 361 samples). Comparison of differentially abundant species between CPE-positive individuals and household controls showed a statistically significant overlap ($\chi^2$ test, $P = 0.0071$) in relation to those we detected to be associated with Enterobacteriaceae in general (Extended Data Fig. 6a). However, the overlap was not significant when comparing CPE-positive to CPE-negative individuals that were previously colonized. This aligns with findings from the original CPE study[10], which noted that individuals recently colonized by CPE are still undergoing microbiome recovery. Nevertheless, we suggest that the overlapping co-excluder and co-colonizer species here discovered (Extended Data Fig. 6b) are not only involved in Enterobacteriaceae colonization as a whole but may also be related to colonization of CPE lineages in particular.

### Strain-specific patterns of Enterobacteriaceae colonization

Our analyses revealed a strong association between the microbiome species composition and Enterobacteriaceae colonization patterns. However, subspecies (that is, strain)-level differences could also be linked to Enterobacteriaceae–microbiome interactions. We therefore investigated strain-specific signatures of 39 gut microbiome species

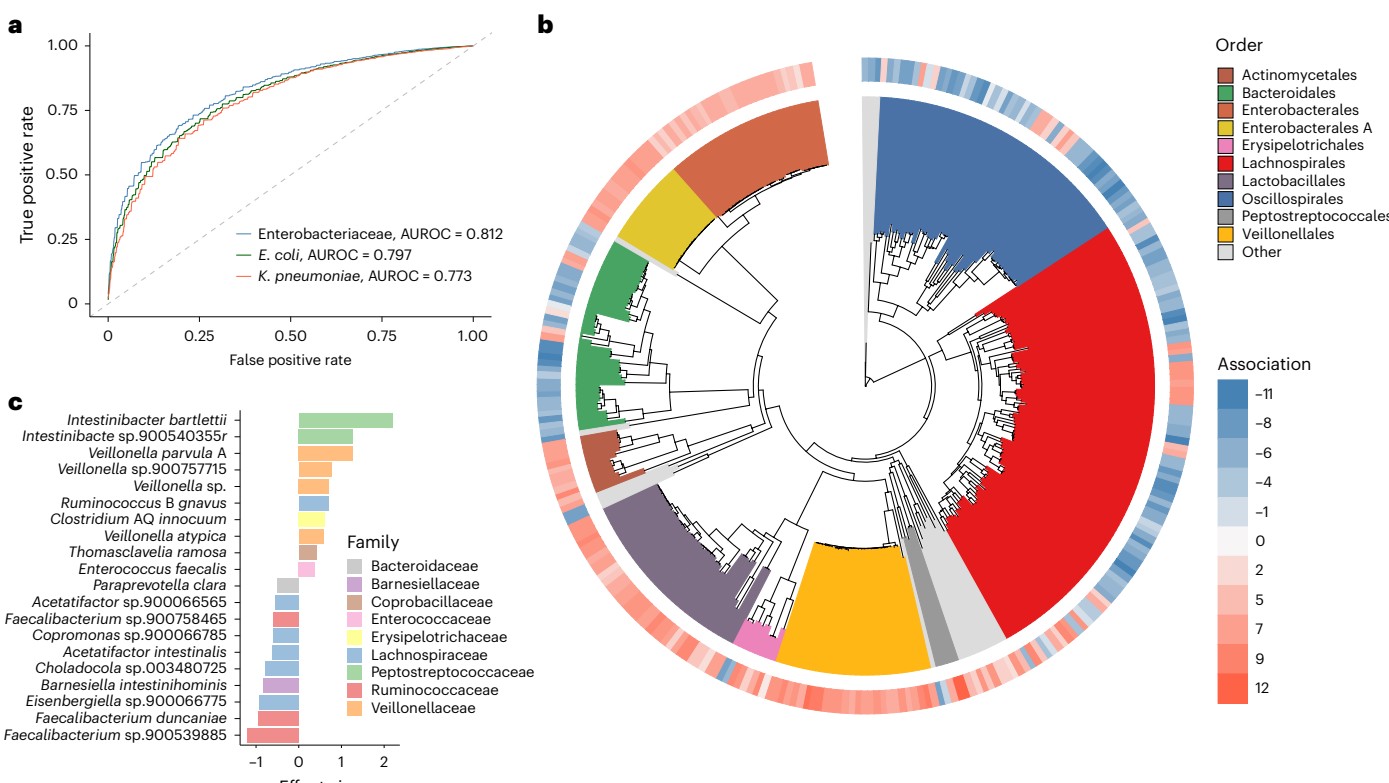

**Fig. 3 | Gut microbiome composition is associated with Enterobacteriaceae colonization and abundance. a**, ROC curve of the machine learning results linking the gut microbiome composition with Enterobacteriaceae, *E. coli* or *K. pneumoniae* colonization status. AUROC values were obtained with gradient boosting applied to the 12,238 human gut metagenomes. **b**, Phylogenetic tree of the 306 bacterial species associated with Enterobacteriaceae colonization and abundance. Clades are coloured according to their affiliated order. Red and blue colours in the outer layer indicate the number of analyses (out of 12) in which each species was classified as a co-excluder (negative) or co-colonizer (positive). A maximum score of 12 denotes that the species was found to be consistently associated with Enterobacteriaecae, *E. coli* and *K. pneumoniae* co-colonization across all 12,238 datasets, as well as the subset of healthy adults. **c**, Top 10 gut microbiome species classified as co-excluders (negative effect size) or co-colonizers (positive effect size) coloured by their family affiliation.

that were identified as either co-colonizers or co-excluders of Enterobacteriaceae among healthy adults, and that were represented by at least 10 genomes with CheckM[25] statistics of >90% completeness and <5% contamination within the UHGG. Using the Unified Human Gastrointestinal Protein (UHGP)[12] catalogue, we characterized the accessory genome (that is, genes detected in <90% of conspecific genomes) of these 39 selected species to identify subspecies populations associated with Enterobacteriaceae colonization. A total of 213 accessory genes were significantly associated with Enterobacteriaceae colonization status (207 positively- and 6 negatively-associated genes; adjusted *P* < 0.05). These were distributed across 15 of the 39 species and overrepresented in functions involved in nucleotide transport and metabolism (Fisher's exact test, adjusted $P = 7.93 \times 10^{-5}$), with the majority belonging to the species *Ruminococcus* B *gnavus* and *Faecalimonas phoceensis* (Extended Data Fig. 7). We further investigated the phylogenetic similarity of strains with the highest number of accessory genes associated with Enterobacteriaceae (top 10% strains; Fig. 4). This revealed a much stronger population structure associated with Enterobacteriaceae co-colonization among *F. phoceensis* strains (permutational multivariate analysis of variance (PERMANOVA), $R^2 = 0.74$, $P < 0.001$) compared with *R. gnavus* genomes (PERMANOVA, $R^2 = 0.02$, $P < 0.001$; Fig. 4). Interestingly, only 2 *Faecalimonas* species among the 307 candidate species were identified as Enterobacteriaceae co-colonizers. In contrast to other genera such as *Veillonella* and *Streptococcus*, which harboured >10 candidate species each, these results suggest that there is a more specific, strain-level association between the diversity of *Faecalimonas* and Enterobacteriaceae co-colonization.

## Co-colonizers are more functionally diverse

Complementing our taxonomic results, we analysed the functional capacity and diversity of the 245 (out of 307) non-Enterobacteriaceae species that consistently showed co-colonizing or co-excluding patterns across datasets and taxa. On the basis of the annotation of protein-coding sequences with KEGG[26], we found that co-colonizers exhibited a greater functional diversity compared with co-excluders (Wilcoxon rank-sum test, $P = 0.0049$; Fig. 5a), which was found to be independent of annotation coverage and genome quality differences (Extended Data Fig. 8a). A previous study showed that metabolic independence drives gut microbial colonization and resilience of disease-associated species, particularly under inflammatory conditions[27]. Here we show that co-colonization of gut bacteria with Enterobacteriaceae is also associated with higher metabolic independence, even among healthy individuals. Statistical analysis of the KEGG Orthologs (KOs) associated with co-colonization or co-exclusion showed that functions involved in drug resistance and DNA regulation (for example, major facilitator superfamily transporter, 16S rRNA methyltransferase and HTH-type transcriptional regulators) are among the functions most strongly linked to co-colonization (Extended Data Fig. 8b and Supplementary Table 3). In contrast, genes related to iron metabolism and transport (for example, rubrerythrin, ferredoxin and ferrous iron transport protein B) show stronger association with colonization resistance. As iron is an essential nutrient for many human pathogens[28], an enrichment of iron-utilizing genes among co-excluders suggests that there is competition for iron availability between Enterobacteriaceae and co-excluder species.

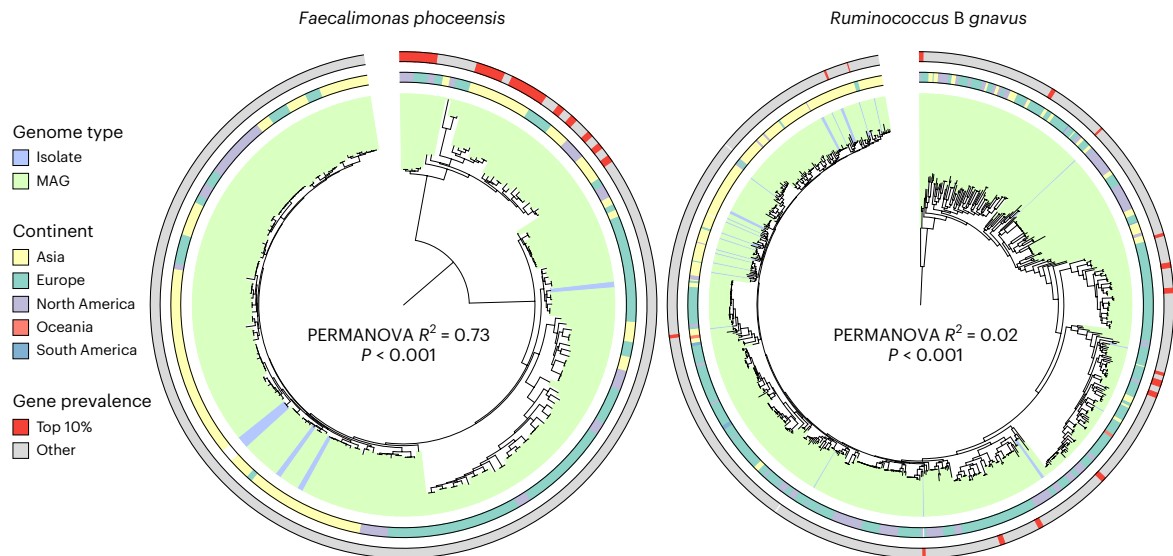

**Fig. 4 | *Faecalimonas phoceensis* exhibits strain-specific co-colonization patterns.** Core-genome phylogenetic tree of 200 *Faecalimonas phoceensis* (left) and 665 *Ruminococcus* B *gnavus* genomes (right). Clades are coloured on the basis of whether the genome is a metagenome-assembled genome (MAG) or an isolate. First outer layer denotes genome geographic origin and the second layer highlights in red those genomes with the highest number of significant accessory genes (top 10%). PERMANOVA was used to relate each species phylogenetic structure (pairwise cophenetic distances) with the number of significant accessory genes. Only genomes with >90% completeness were included in the analysis.

On a broader level, we characterized the main functional categories (Clustered Orthologs Groups, COGs) associated with Enterobacteriaceae co-colonization and co-exclusion (Fig. 5b). In general, functions involved in metabolism (for example, amino acids, nucleotides and inorganic ions) were enriched among co-colonizers (Fisher's exact test, adjusted $P < 0.05$), which further supports the hypothesis that co-colonizers exhibit greater metabolic independence. On the other hand, co-excluders encoded a higher number of genes involved in signal transduction mechanisms, which includes functions such as sporulation, motility and quorum sensing. We also identified more genes with unknown functions among co-excluders, suggesting that they may carry uncharacterized mechanisms with potential roles in Enterobacteriaceae colonization resistance. These results were consistent even when only considering species from the Bacillota phylum, which contains the highest number of both co-excluders and co-colonizers (Extended Data Fig. 8c).

We further investigated the distribution of specialized primary metabolic pathways associated with Enterobacteriaceae colonization patterns using the gutSMASH[29] algorithm. We found that co-excluders were primarily enriched in metabolic gene clusters involved in the production of the three major SCFAs (acetate, propionate and butyrate), as well as in other pathways involving the Rnf complex and 2-hydroxyglutaryl-CoA dehydratases (Fig. 5c). These results reinforce the genus-wide signal we observed for *Faecalibacterium*, one of the notable short-chain fatty acid producers in the gut. In addition, the Rnf complex signature further supports the importance of iron among co-excluders, as the Rnf complex consists of a ferredoxin:NAD$^+$ oxidoreductase involved in energy production in anaerobic bacteria[30]. These analyses reveal that species co-occurring with Enterobacteriaceae are more functionally diverse, with iron metabolism and SCFAs potentially playing important roles in regulating gut environmental conditions and modulating Enterobacteriaceae colonization and abundance.

### Gut colonization is largely driven by habitat filtering

To investigate the relationship between co-colonization patterns and interspecies metabolic interactions between Enterobacteriaceae species and the rest of the gut microbiome, we generated genome-scale metabolic models[31] of all candidate co-colonizers and co-excluders (only considering those with a consistent signal across datasets and taxa), together with all Enterobacteriaceae species from the UHGG detected at >1% prevalence ($n = 282$). Using a phylogenetically adjusted quantification method[32], we calculated metabolic competition and complementarity indices between all pairwise Enterobacteriaceae–microbiome combinations. Metabolic competition was calculated on the basis of the overlap between two given metabolic networks, while metabolic complementarity measured the potential of one species to utilize the metabolic output of another. Values were distributed across two main clusters based on high or low competition/complementarity indices (Extended Data Fig. 9a). However, within each cluster, there was a significant negative correlation between metabolic competition and complementarity (Cluster1: Pearson's $r = -0.85$, $P < 0.0001$; Cluster2: Pearson's $r = -0.20$, $P < 0.0001$). We combined both parameters to estimate a metabolic distance score and observed that co-colonizers showed a lower metabolic distance to Enterobacteriaceae species compared with co-excluders (Wilcoxon rank-sum test, $P < 0.0001$; Fig. 5d and Extended Data Fig. 9b). These results indicate that habitat filtering, a process that favours the coexistence of functionally similar species, is probably the main driver of colonization success and microbiome assembly among Enterobacteriaceae-associated species. In addition, metabolic comparisons were made both within and between the groups of co-excluders and co-colonizers. These analyses revealed that within-group metabolic distances were smaller than between-group differences (Wilcoxon rank-sum test, $P < 0.0001$; Extended Data Fig. 9c), indicating shared niche preference. Importantly, results were consistent even when simulating metabolic models under various media compositions reflecting differences in diets (Extended Data Fig. 9d). This supports a previous study showing that gut microbiome species with similar nutritional requirements tend to co-occur across individuals[33].

To further explore functional differences between co-excluders and co-colonizers inferred from metabolic models, we compared the number and types of metabolite predicted from uptake and secretion fluxes within each species population. By simulating the models under a rich gut medium (M3)[34], we observed that the number of metabolites predicted to be secreted was significantly higher among co-colonizers (Wilcoxon rank-sum test, $P < 0.0001$, Extended Data Fig. 10a), further

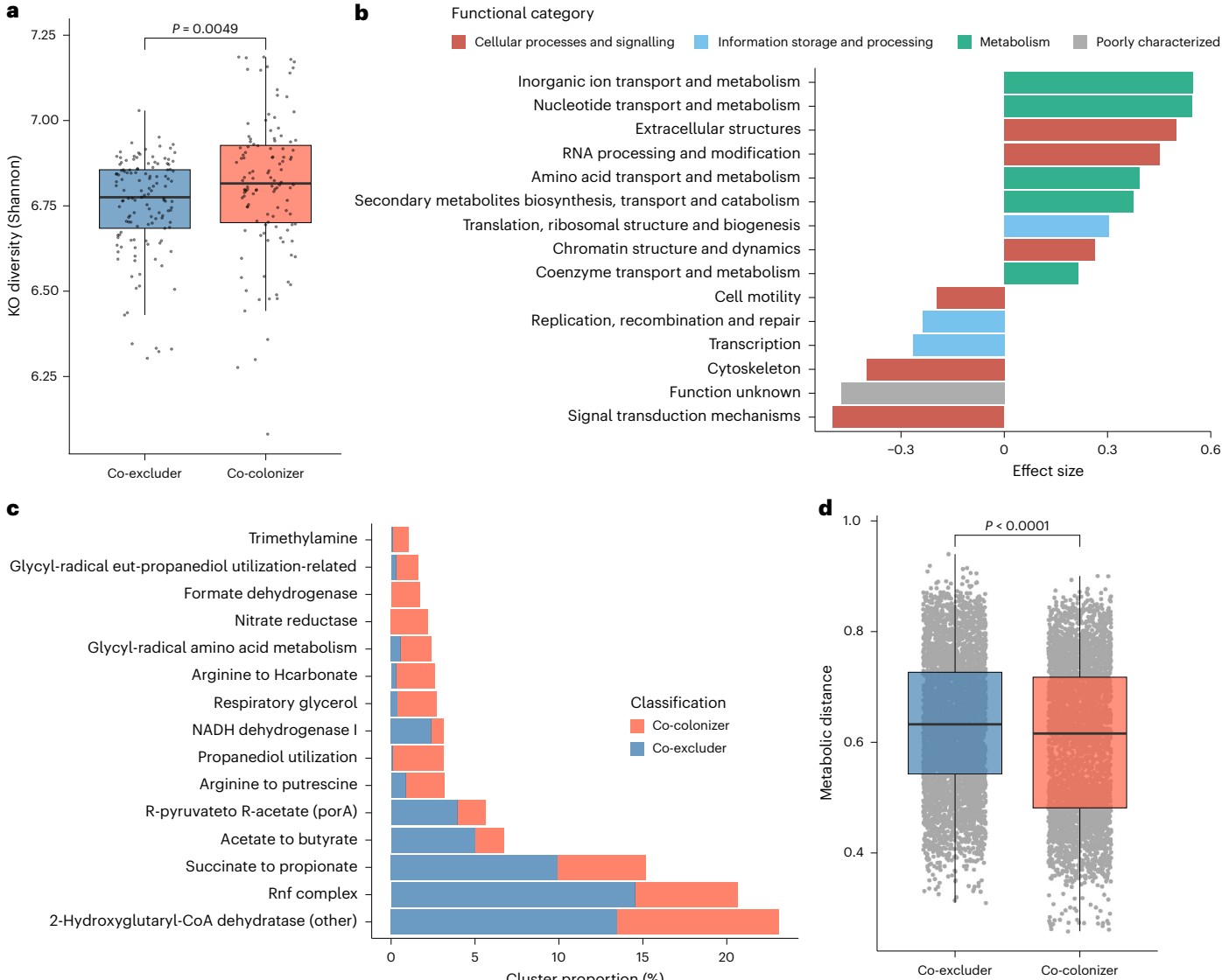

**Fig. 5 | Functional differences between co-excluders and co-colonizers.**
**a**, Distribution of Shannon diversity values obtained among co-excluders ($n$ = 129) and co-colonizers ($n$ = 116), which did not belong to the Enterobacteriaceae family, based on the pattern of KOs detected per genome. Exact $P$ values were calculated with a two-sided Wilcoxon rank-sum test. **b**, COG functional categories significantly associated with co-colonizers (positive effect size) or co-excluders (negative effect size). **c**, Primary metabolic pathways detected with gutSMASH differentially abundant between co-excluders and co-colonizers. **d**, Pairwise metabolic distances between co-excluders or co-colonizers compared to all Enterobacteriaceae species detected at >1% prevalence (co-excluders: $n$ = 4,773 comparisons; co-colonizers: $n$ = 4,292 comparisons). $P$ values were calculated with a two-sided Wilcoxon rank-sum test. In **a** and **d**, box lengths represent the IQR of the data, the central line represents the median, and the whiskers depict the lowest and highest values within 1.5× the IQR of the first and third quartiles, respectively.

supporting that higher functional diversity and versatility is associated with Enterobacteriaceae co-colonization. Statistical analyses identified 7 and 77 candidate metabolites from uptake or secretion fluxes, respectively, as differentially abundant between co-colonizers and co-excluders (Extended Data Fig. 10b). With regards to estimated uptake fluxes, L-serine and indole were the most significant metabolites enriched in co-colonizers and co-excluders, respectively. Interestingly, dietary L-serine has been previously described to provide a competitive fitness advantage to Enterobacteriaceae under inflammatory conditions[35], while indole has been shown to alleviate intestinal inflammation through modulation of the gut microbiome composition[36]. Indole is also recognized as a signalling molecule among indole-producing bacteria, such as *E. coli*[37]. Therefore, a higher uptake of indole among co-excluders could impair intercellular signal communication of Enterobacteriaceae. In terms of secretion, we observed most notably an overrepresentation of the metabolites

undecaprenyl phosphate/undecaprenyl-diphosphatase and thymine among co-excluders, concomitant with a higher secretion of oxidized glutathione and β-alanine among co-colonizers. Undecaprenyl phosphate is involved in the biogenesis of the bacterial cell wall[38], while thymine is essential for DNA synthesis, repair and bacterial growth. In contrast, detection of oxidized glutathione among co-colonizers may be indicative of adaptation to an environment with higher oxygen tension and oxidative stress. Lastly, β-alanine, which we found to be overrepresented among co-colonizers, was previously identified as significantly increased in Crohn's disease patients[39]. Overall, these results highlight metabolic differences between co-excluders and co-colonizing species that may reflect differences in colonization and adaptation to distinct gut niches. This further supports that modulation of the gut environment, for instance through diet, may affect susceptibility to Enterobacteriaceae colonization.

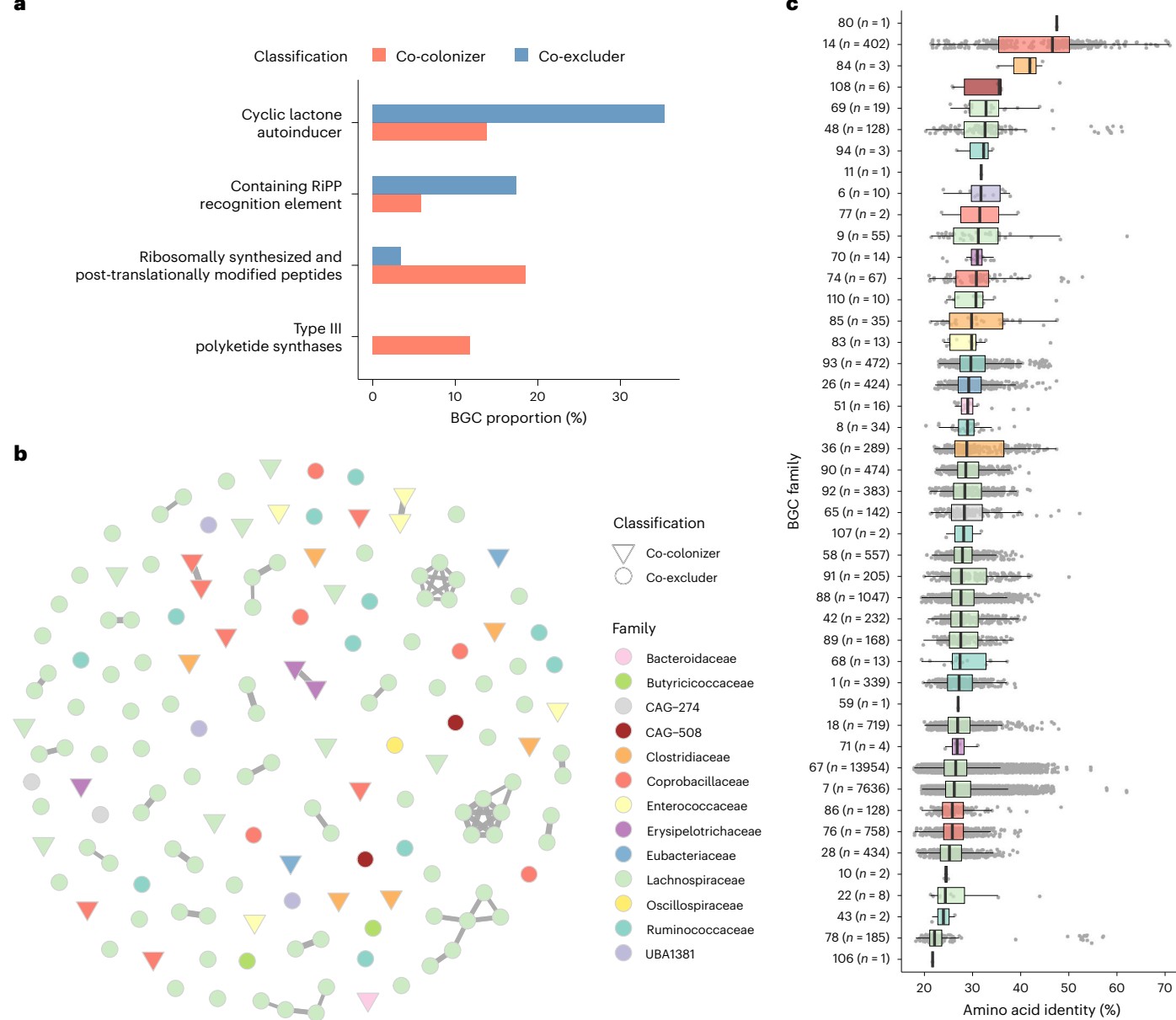

**Fig. 6 | Co-excluders harbour biosynthetic gene clusters involved in quorum sensing. a**, BGCs detected with antiSMASH that were found to be overrepresented among co-excluders or co-colonizers (two-sided Fisher's exact test, adjusted $P < 0.05$). **b**, Network of all cyclic lactone autoinducer BGCs detected among co-colonizers and co-excluders. BGC nodes are linked if they share >50% nucleotide identity over >50% alignment coverage. **c**, Distribution of amino acid identity values obtained by comparing each autoinducer BGC family against the MIBiG database ($n$ values indicated in parenthesis next to each BGC family represent the total number of alignments against the MIBiG database). Box lengths represent the IQR of the data, the central line represents the median, and the whiskers depict the lowest and highest values within 1.5× the IQR of the first and third quartiles, respectively.

## Co-excluders encode gene families involved in quorum sensing

As the production of secondary metabolites can influence bacterial fitness and interspecies ecological interactions, we performed a dedicated analysis of biosynthetic gene clusters (BGCs) among Enterobacteriaceae co-colonizers and co-excluders using the antiSMASH[40] prediction tool. The BGCs identified as most significantly overrepresented among co-excluders belonged to a class of cyclic lactone autoinducers (Fisher's exact test, adjusted $P < 0.05$; Fig. 6a). Autoinducers represent signalling molecules that play a role in bacterial communication through quorum sensing[41]. This is in line with the COG results which showed an enrichment of signal transduction mechanisms among co-excluders. By investigating the taxonomic distribution of all 147 autoinducer BGCs detected, we find that the majority were harboured by species from the *Lachnospiraceae* family (97/147, 66%). Further grouping of the BGCs by their genetic similarity (>50% nucleotide identity over >50% coverage; Supplementary Table 4) revealed 110 unique BGC families, segregated between co-excluders and co-colonizers (Fig. 6b). We compared all genes from each family to experimentally characterized BGCs in the Minimum Information about a Biosynthetic Gene Cluster (MIBiG) database[42] and found an overall low amino acid identity to known clusters (interquartile range, IQR = 25–30%; Fig. 6c), suggesting that these autoinducer BGCs represent uncharacterized sequences. Quorum sensing molecules have been previously implicated as a means for members of the gut microbiome to provide colonization resistance against external pathogens[43]. Therefore, an enrichment of these autoinducer BGCs among co-excluders

indicates that they could play a role in controlling the colonization and abundance of Enterobacteriaceae in the human gut.

## Discussion

We performed a large, global characterization of gut microbiome signatures linked to Enterobacteriaceae colonization and abundance, revealing taxonomic and functional shifts related to co-colonization and co-exclusion. In addition to confirming previous associations (for example, negative signal of Enterobacteriaceae with SCFA producers) in a diverse large-scale cohort, our study provides several important insights. First, we reveal significant and consistent microbiome differences associated with Enterobacteriaceae colonization across health states and geographic regions, uncovering a large uncharacterized subspecies diversity of *E. coli* among healthy adults in Africa. Moreover, our findings suggest that species from the *Faecalibacterium* genus beyond the well-established *F. prausnitzii* may play a critical role in colonization resistance against Enterobacteriaceae. We also identified notable co-colonization patterns involving underexplored taxa such as *Intestinibacter* and *F. phoceensis* that may underlie biological mechanisms linked to Enterobacteriaceae outgrowth. Finally, we discovered that co-excluder species harbour a range of uncharacterized BGCs involved in quorum sensing that could be modulating Enterobacteriaceae abundance.

Despite representing a large metagenomic investigation of Enterobacteriaceae–microbiome dynamics, our study has some inherent limitations. Due to the compositional nature of sequencing data, we were unable to differentiate between absolute and relative abundance estimates. Metagenomic samples may show varying relative abundances of Enterobacteriaceae, yet reflect similar absolute colonization levels. In addition, it remains unclear whether co-excluders are a cause or consequence of reduced Enterobacteriaceae levels, as species such as *E. coli* could potentially inhibit co-excluder growth through mechanisms such as antimicrobial production. Lastly, although we analysed metagenomic data from 45 countries, many regions of the world, particularly South America and Africa, remain undersampled.

A recent study of two large-scale population cohorts identified that baseline gut microbiome composition is an important risk factor for infection-related hospital admission[44]. In line with our results, lower relative abundances of *Veillonella* and higher abundances of butyrate producers were associated with protection against hospitalization. These findings suggest that some of the associations here described might not only be correlated with Enterobacteriaceae colonization, but could also be predictive of distinct health outcomes and overall infection risk.

Previous studies in animal models[45–49] and perturbed microbiomes[50] have described competitive interactions among Enterobacteriaceae species, including between *E. coli* strains and other Enterobacteriaceae species or between members of the *Klebsiella* genus specifically. However, in our study we detected a significant co-occurrence of species that are predicted to metabolically compete, including those within the Enterobacteriaceae family. Therefore, our results suggest that habitat filtering has a stronger effect on Enterobacteriaceae colonization success in the human gut compared with that observed in animal models, where the impact of direct interspecies competition may be more pronounced. In addition, most of our findings were inferred from healthy populations, which exhibit substantial microbiome differences compared with disturbed microbiomes such as those undergoing transplantation or antibiotic treatment. Altogether, these data indicate that gut environmental conditions, for instance mediated by diet, play a key role in the risk of Enterobacteriaceae outgrowth. Designing large-scale longitudinal and/or interventional studies combining metagenomics, metatranscriptomics and metabolomics to test the role of diet, medication and the environment in Enterobacteriaceae colonization and abundance represent promising future directions.

In summary, our research provides important insights into the biological role of the human gut microbiome in the colonization success of Enterobacteriaceae in the human gut. With the global rise of multidrug-resistant Enterobacteriaceae and the negative health outcomes associated with Enterobacteriaceae outgrowth, our findings could guide future research towards developing microbiome-based therapeutic strategies.

## Methods

### Human gut metagenomic datasets

We compiled 12,238 human gut metagenomic samples available in the European Nucleotide Archive (ENA)[51], encompassing 65 different studies from 45 countries (Supplementary Table 1). Samples were selected on the basis of the following criteria: (1) containing at least 500,000 paired-end metagenomic reads; (2) with available metadata on health state, age group and country of origin; (3) from individuals with no diagnosed acute infections; and (4) no reported antibiotic usage in the previous month. Metagenomic datasets were first downloaded from ENA using fastq-dl v.2.0.4 (https://github.com/rpetit3/fastq-dl) and further quality-filtered with TrimGalore (v.0.6.0)[52]. Human contamination was removed by aligning the reads using BWA MEM (v.0.7.16a-r1181)[53] against human genome GRCh38. The custom pipeline used for downloading and quality-filtering the human gut metagenomic samples can be found in GitHub at https://github.com/alexmsalmeida/metagen-fetch.

### Species prevalence and abundance

Quality-filtered metagenomes were mapped and quantified against the UHGG (v.1.0) catalogue[12]. Before read mapping, genomes from the UHGG were curated to filter those matching all of the following criteria: (1) singletons (that is, species represented by only one genome); (2) <90% completeness based on CheckM (v.1.0.11)[25]; and (3) classified by GUNC (v.1.0.3)[54] as chimaeric ('clade_separation_score' >0.45, 'contamination_portion' >0.05 and 'reference_representation_score' >0.5). This removed 32 species, resulting in a total of 4,612 species representatives. In addition, UHGG species representatives were taxonomically reclassified using GTDB-Tk (v.2.3.2)[55] (database release 214). The final curated database was indexed using BWA v.0.7.16a-r1181 ('bwa index'), and metagenomic sequence reads were mapped using 'bwa mem'. Aligned reads were filtered using Samtools (v.1.9)[56] to solely keep alignments where >60% of the read matched with >90% identity against any species representative. Breadth of coverage, depth of coverage, total read counts and counts of uniquely mapped reads were calculated per sample for each species in the reference database. In addition, to account for differences in sample sequencing depth, an expected breadth of coverage ($E$) per sample per species was calculated using a previously established formula[57]:

$$E = 1 - e^{-0.883D} \tag{1}$$

where $D$ corresponds to the average depth of coverage of the genome. Each species was considered present in a sample if the ratio between the breadth of coverage and the expected breadth of coverage was >30% (given that genomes from the UHGG were originally clustered using a 30% aligned fraction) and if the breadth of coverage was >5% (to account for the presence of metagenome-assembled genomes, MAGs, with up to 5% contamination). Read counts were transformed to 0 when a species was considered absent. Parameters of genome coverage and coverage ratio were validated with the ZymoBIOMICS Microbial Community Standard. In addition, we generated synthetic metagenomic communities to establish the detection limit of our metagenomic analysis approach. Briefly, metagenomes containing the 50 most prevalent species detected in our dataset at equal relative abundances were simulated with 'wgsim' in Samtools[56]. Thereafter, we spiked-in one Enterobacteriaceae species at various known relative

abundances (from 0.0001% to 1% at 8 intervals) and processed the sample with the mapping approach described above. Each analysis was repeated for five Enterobacteriaceae species representing the top five most prevalent genera (*E. coli*, *K. pneumoniae*, *E. hormaechei*, *Citrobacter freundii* and *Kluyvera ascorbata*) and three levels of sequencing depth based on the distribution observed for our dataset: low depth (first quartile) = 13 million reads; medium depth (median) = 31 million reads; high depth (third quartile) = 51 million reads. This represented a total of 120 synthetic communities. The detection limit was defined as the minimum relative abundance at which all five species were detected. The custom read mapping and species quantification workflow is publicly available as a snakemake[58] pipeline in GitHub at https://github.com/alexmsalmeida/metamap.

To account for study-specific batch effects, species abundances were subsequently corrected using a conditional quantile algorithm implemented in ConQuR (v.1.2.0)[59]. Each of the 65 studies was tested as a reference in the batch correction process, and we ultimately used study ERP111320 as the final reference, which yielded the lowest study effect size ($R^2$ = 0.068) based on a PERMANOVA.

## Microbiome community differences

To evaluate the potential of the gut microbial community structure in classifying the colonization status of Enterobacteriaceae across all 12,238 metagenomes, we tested three supervised machine learning algorithms (ridge regression, random forest and gradient boosting) using a custom workflow (https://github.com/alexmsalmeida/ml-microbiome) derived from the Mikropml[60] R package. The abundance of each filtered microbiome species was transformed into centred-log ratios (clr) and used as features in each machine learning model. Features were further pre-processed to exclude those with prevalence <1% and with zero variance. Model training and hyperparameter tuning was performed on 80% of the data using a 5-fold cross-validation, while the other 20% was used for testing with the best hyperparameter setting. The whole procedure was then repeated 10 times with independent seeds. We performed the analysis using, as outcome variable, the presence or absence of any Enterobacteriaceae species, or of *E. coli* and *K. pneumoniae* in particular. The study source was used as a grouping factor to ensure that samples from the same study were kept together in either the training or testing dataset. Model performance was evaluated using the AUROC.

Diversity estimates were inferred from the species abundance data. Alpha diversity was calculated using the Shannon index in the vegan[61] R package, on the basis of the number of unique read counts mapped to each species per sample. Beta diversity between samples was estimated from the Aitchison distance of the clr-transformed species abundances. Correlation between alpha diversity and Enterobacteriaceae abundance was assessed with the Pearson's coefficient of determination ($R^2$), while differences in pairwise beta diversity were calculated using a two-sided Wilcoxon rank-sum test. To further account for differences in sequencing depth, results were confirmed by rarefying the number of mapped reads to 500,000.

## Identification of candidate species

To identify differentially abundant microbiome species associated with the presence/absence of Enterobacteriaceae (colonization status), we utilized a combination of generalized (ALDEx2 (v.1.32.0)[18]) and mixed-effects (MaAsLin2 (v.1.14.1)[19]) models. To be able to appropriately control for potential confounders, the differential analyses spanned two datasets: (1) a complete collection of all 12,238 metagenomes with metadata including age group, health state (that is, specific disease name or 'Healthy'), continent and study, which were incorporated as model covariates; and (2) a subset of 5,128 metagenomes exclusively obtained from healthy adults, controlled for both study and continent. In both datasets, read counts were included as an additional covariate to account for differences in sequencing depth between

samples. Analyses were conducted using Enterobacteriaceae, *E. coli* or *K. pneumoniae* presence/absence as outcome variables. For ALDEx2, we used the 'aldex.clr' function followed by 'aldex.glm' to model abundance differences after a centred-log ratio transformation. For MaAsLin2, we used a log transformation combined with relative abundance (total sum scaling) normalization and accounting for genome length. A minimum prevalence threshold of 1% was used to filter out rare species from both analyses. *P* values were corrected for multiple testing using the Benjamini–Hochberg method and only species with a false discovery rate (FDR) < 5% were considered significant.

We used FastSpar (v.1.0)[21] (a C++ implementation of SparCC[20]) to generate correlation networks between Enterobacteriaceae species abundances and the abundances of other microbiome members found at >1% prevalence. As above, batch-corrected read counts were used as species abundances. Exact *P* values were calculated from 1,000 bootstrap correlations and corrected for multiple testing using the Benjamini–Hochberg method. Only correlations with an FDR < 5% were kept. To select the final candidate species, only those that were statistically significant across ALDEx2, MaAsLin2 and FastSpar were chosen. On the basis of the overall direction of the signal (positive or negative), species were classified as either Enterobacteriaceae co-colonizers or co-excluders.

Results were further compared with the dataset of ref. 10 that specifically investigated the longitudinal dynamics of carbapenemase-producing Enterobacteriaceae (CPE). A total of 361 samples were processed from study ERP133829 and mapped to the UHGG as described above. To identify differentially abundant species while accounting for the longitudinal study design, we used the mixed-effects modelling implemented in MaAsLin2. The CPE status (CPE-positive, CPE-negative control or CPE-negative index) was used as a fixed effect, and the individual participant was used as a random effect. Samples from individuals who received antibiotics or were hospitalized since their last visit were excluded from the analysis. As above, the minimum prevalence threshold was set at 1% and statistical significance was determined with an FDR < 5%. The overlap with the species identified as Enterobacteriaceae co-excluders or co-colonizers was inferred with a $\chi^2$ test using four estimates: (1) number of species associated with CPE and Enterobacteriaceae; (2) number of species associated only with CPE; (3) number of species associated with Enterobacteriaceae but not CPE; and (4) number of species not associated with either.

## Strain-specific colonization patterns

A metagenomic multilocus sequence typing (metaMLST[14]) analysis was performed to characterize *E. coli* subspecies diversity. Metagenomic reads from 5,128 healthy adults were aligned using bowtie2 (v.2.5.3)[62] (option '−very-sensitive-local') against the 7 housekeeping genes from the Achtman *E. coli* MLST scheme (containing 13,253 STs as of April 2024). Mapping results were further processed using the 'metamlst.py' (option '−min_accuracy 0.5') and 'metamlst-merge.py' (option '-z 10') Python scripts in the metaMLST (v.1.2.3) tool. Results were further filtered to only include STs with a confidence score >90. A minimum spanning tree was built using the igraph[63] R package on the basis of the Euclidean distances of the ST allelic profiles detected.

We further explored the strain-specific genetic diversity of species identified as either co-colonizers or co-excluders of Enterobacteriaceae among healthy adult samples. To account for differences in genome quality and overall number of strains, only gut species with at least 10 genomes with >90% completeness and <5% contamination represented within the UHGG database were chosen. We extracted all accessory genes (<90% prevalence) exclusive to each candidate species from the UHGP catalogue[12] clustered at 90% amino acid identity (UHGP-90). In parallel, we generated metagenome assemblies of the 5,128 samples from healthy adults using MEGAHIT (v.1.2.9)[64] (option '−min-contig-len 500'), followed by protein prediction with Prodigal (v.2.6.3)[65] (option

'-p meta'). Thereafter, we used DIAMOND (v.2.1.8)[66] (function 'blastp') to align all extracted UHGP genes against the predicted proteins of the healthy adult metagenomes (using thresholds of 90% amino acid identity and 80% coverage of the shortest sequence). This resulted in a binary (presence/absence) matrix of all genes across all metagenomic samples. Subsequently, a generalized linear model (glm R function, family = 'binomial') was employed to investigate the association of the accessory genes with Enterobacteriaceae colonization status (presence/absence), using 'Study' as a covariate. *P* values were corrected for multiple testing using the Benjamini–Hochberg method and only genes with an FDR < 5% were considered significant.

On the basis of the number of significant genes identified, two species (*Ruminococcus* B *gnavus* and *Faecalimonas phoceensis*) were chosen for further phylogenetic analysis. After extracting all conspecific genomes (>90% complete) available in the UHGG, we used Panaroo (v.1.3.3)[67] to perform a species-specific core genome alignment (options: '−clean-mode strict −remove-invalid-genes -c 0.9 -f 0.5 −merge_paralogs −core_threshold 0.9 -a core'). FastTree (v.2.1.11)[68] was subsequently used to build the phylogenetic tree, which was visualized with the interactive Tree of Life (iTOL) (v.6)[69] online tool. A PERMANOVA was used to assess the association between the cophenetic distances in the phylogeny and the number of significant accessory genes per genome.

### Genome functional analyses

Functional analyses were performed on candidate microbiomes with a consistent positive or negative signal in relation to Enterobacteriaceae colonization and abundance. First, protein-coding sequences were predicted from the genome assemblies using Prokka (v.1.14.16)[70]. Thereafter, we used a custom workflow (https://github.com/alexmsalmeida/genofan) to comprehensively annotate each genome using eggNOG-mapper (v.2.1.3)[71], dbCAN2 (v.2.0.11)[72], KOFam[26] (release 2021-11), gutSMASH (v.1.0)[29] and antiSMASH (v.6.0.1)[40]. The diversity of KEGG Orthologs (KOs) among co-excluders and co-colonizers was derived from the KOFam results and further assessed using the Shannon diversity index implemented in the vegan R package. A generalized linear model was used to identify KOs significantly associated with co-exclusion or co-colonization (glm R function, family = 'binomial') while controlling for genome type (MAG or isolate) as a covariate. COGs were extracted from the eggNOG results and compared using a two-tailed Wilcoxon rank-sum test. Lastly, the proportions of gene clusters, retrieved using gutSMASH and antiSMASH, between co-colonizers and co-excluders were evaluated using Fisher's exact test. In all statistical analyses, exact *P* values were corrected for multiple testing and filtered on the basis of an FDR < 5%. BGCs belonging to the class of cyclic lactone autoinducers were further analysed using an all-against-all blast analysis, and a network was built using the igraph R package, linking nodes (BGCs) with >50 nucleotide identity over >50% coverage. Sequence identity to known clusters from the MIBiG database was derived using the 'KnownClusterBlast' algorithm implemented in antiSMASH.

Genome-scale metabolic models were generated for each genome using CarveMe (v.1.5.2)[31] with default parameters and gap-filled with gut-specific rich media (M3)[34]. From the reconstructed models, we estimated metabolic competition and complementarity indices between all candidate species and all Enterobacteriaceae species detected at >1% prevalence using PhyloMint (v.0.1.0)[32]. Given the negative correlation observed between both measures, we also calculated a combined metabolic distance score (1 − (competition index − complementarity index)). Results were further confirmed using flux balance analysis simulation of metabolic models under defined gut media (M1)[34] supplemented with various diet-based media available in the Virtual Metabolic Human database[73]. The number and types of metabolites predicted from secreted or uptake fluxes among co-colonizers and co-excluders were derived using the COBRApy (v.0.29)[74] Python package and further

compared using a generalized linear model as described above for the identification of candidate KOs.

### Reporting summary

Further information on research design is available in the Nature Portfolio Reporting Summary linked to this article.

## Data availability

All the metagenomic datasets used in this study are publicly available in the European Nucleotide Archive (see Supplementary Table 1 for all associated accession codes). The sequence databases used were retrieved from the Unified Human Gastrointestinal Genome (UHGG) catalog v.1.0 and the Unified Human Gastrointestinal Protein (UHGP-90) catalog v.1.0. Abundance data estimated for the UHGG species and all metagenomic samples here included are available in figshare at https://doi.org/10.6084/m9.figshare.27044341.v1 (ref. 75). FASTA files of the BGCs detected with antiSMASH for all co-excluders and co-colonizers can be accessed in figshare at https://doi.org/10.6084/m9.figshare.27044335.v1 (ref. 76). Accession code of the human reference genome used for decontamination (GRCh38) is GCA_000001405.15.

## Code availability

Custom scripts and pipelines used in this work are publicly available in GitHub at https://github.com/microfundiv-lab/EnteroEco (ref. 77).

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

## Acknowledgements

We thank J. Parkhill and all members of the Microbiome Function and Diversity group for helpful feedback and suggestions; and all authors who collected and made the gut metagenomic datasets used in this study publicly available. Funding was provided by a Career Development Award from the Medical Research Council (MR/W016184/1) to A.A.; and 2021.02791.CEECIND (Fundação para a Ciência e Tecnologia) to A.S.A. The funders had no role in study design, data collection and analysis, decision to publish or preparation of the manuscript.

## Author contributions

Q.Y. and A.A. performed the metagenomic analyses and wrote the paper. A.C.d.S. helped in the curation of the sample metadata. F.Z. and K.R.P. assisted with the genome-scale metabolic modelling. A.S.A. processed and provided the mock community samples. A.A. supervised the work and provided funding. All authors read, edited and approved the paper.

## Competing interests

The authors declare no competing interests.

## Additional information

**Extended data** is available for this paper at https://doi.org/10.1038/s41564-024-01912-6.

**Correspondence and requests for materials** should be addressed to Alexandre Almeida.

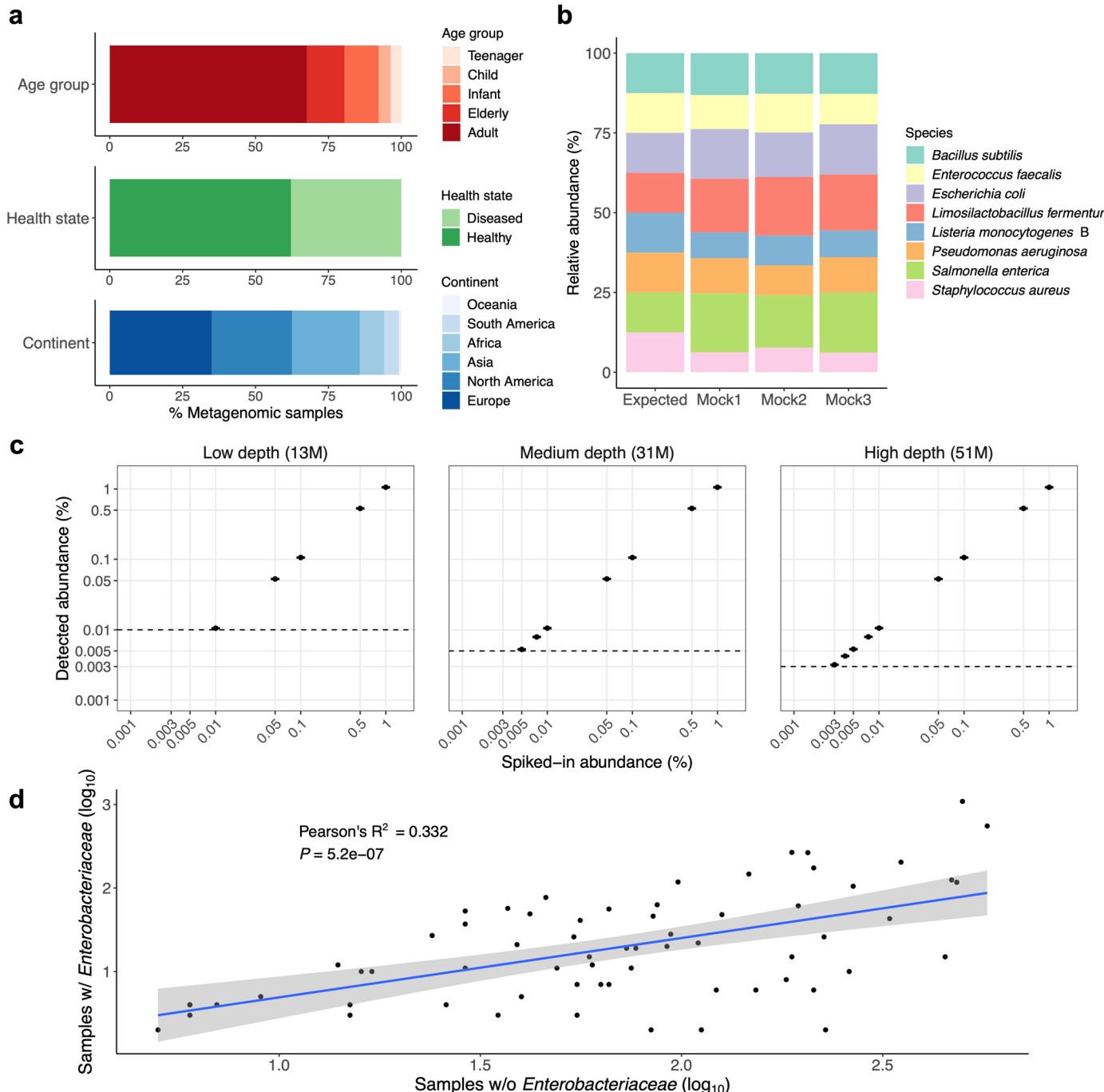

**Extended Data Fig. 1 | Sample distribution and mapping quality control.**
**a**, Distribution of age groups, health states and continents of the 12,238 gut metagenomic samples. **b**, Comparison of taxonomic profiles and abundances of three mock community samples in relation to their expected proportions, estimated using the read mapping filtering parameters used in this study. **c**, Detection limit of our metagenomic approach evaluated with 120 synthetic metagenomics consisting of the top 50 most prevalent gut species and one

Enterobacteriaceae species at a defined abundance across three levels of sequencing depth. Horizontal dashed line represents the minimum relative abundance at which the five Enterobacteriaceae species tested were detected. Abundance values are log-scaled. **d**, Two-sided Pearson correlation between the number of samples with or without Enterobacteriaceae across the 65 studies. Error band represents the 95% confidence interval.

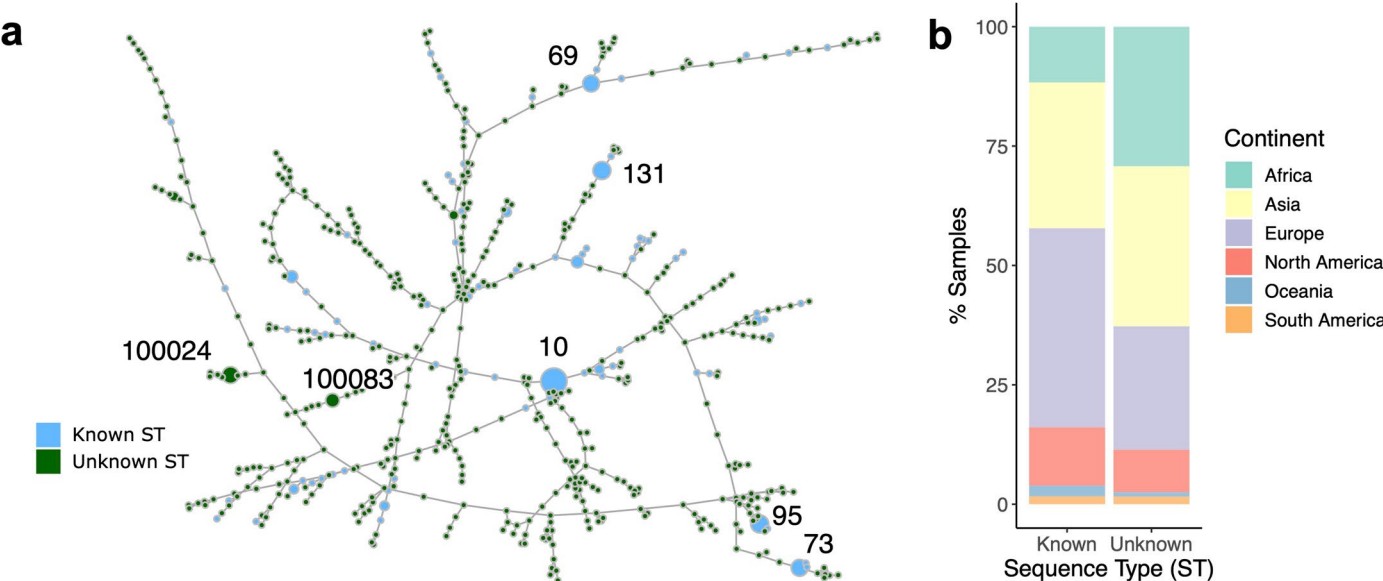

**Extended Data Fig. 2 | Strain diversity of *Escherichia coli* in the human gut microbiome among healthy adults. a**, Minimum spanning tree of the *E. coli* sequence types (STs) detected across 5,128 human gut metagenomes from healthy adults. The most prevalent STs are labelled next to their respective nodes (ST100024 and ST100083 represent unknown STs). **b**, Geographical distribution of samples containing known or unknown STs.

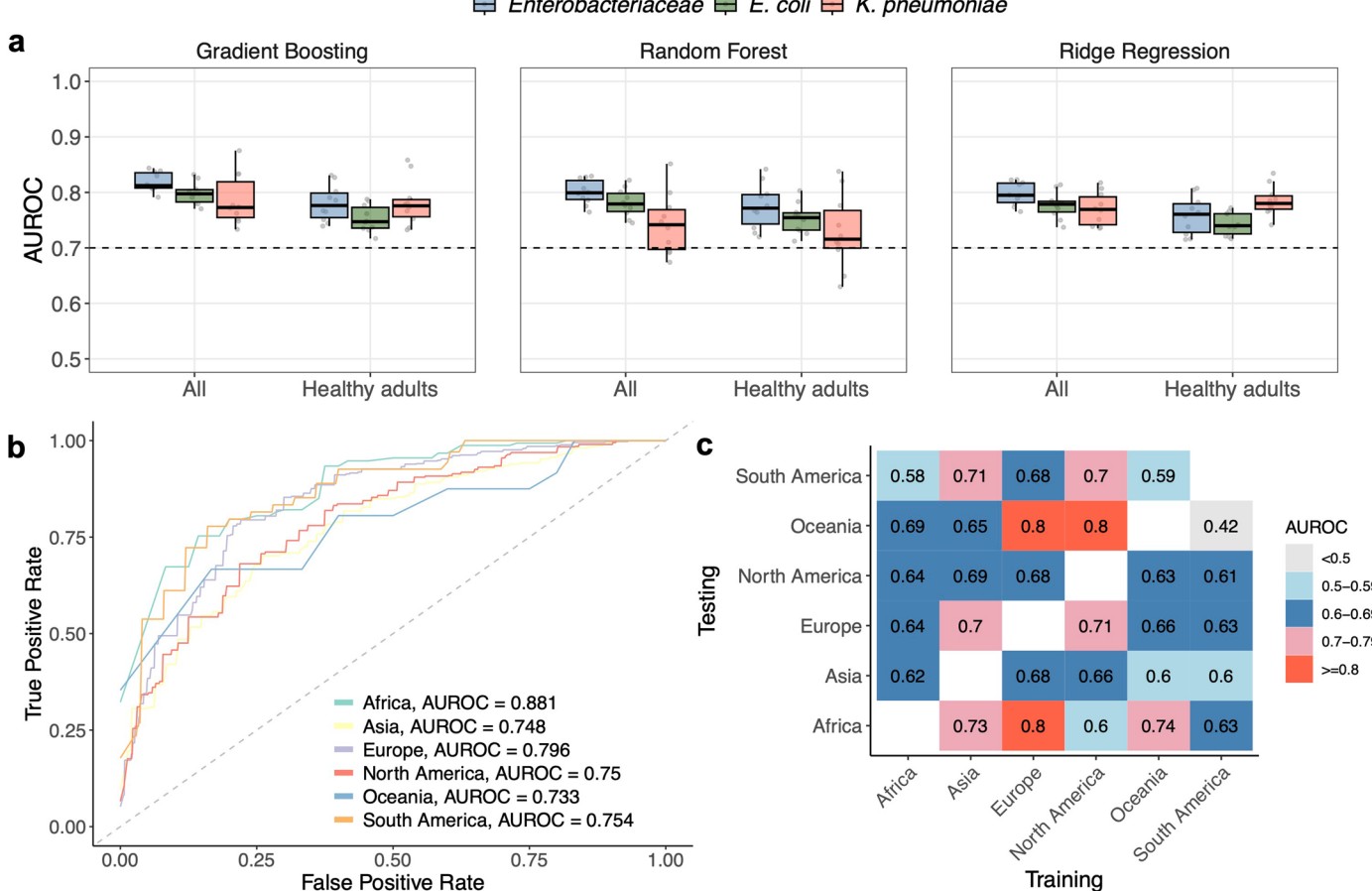

**Extended Data Fig. 3 | Machine learning models to classify Enterobacteriaceae colonization status. a**, Area Under the ROC Curve (AUROC) performance results of different machine learning methods, datasets and outcome variables (taxa) relating the gut microbiome composition with Enterobacteriaceae colonization status (*n* = 10 independent seeds per analysis). Box lengths represent the IQR of the data, the central line represents the median value, and the whiskers depict the lowest and highest values within 1.5 times the IQR of the first and third quartiles, respectively. **b**, ROC curve of the machine learning results linking the

gut microbiome composition with Enterobacteriaceae status. AUROC values represent the median of gradient boosting models across 10 independent seeds, stratified by continent and only considering samples from healthy adults **c**, All-against-all performance results comparing models trained and tested using microbiome samples across different continents. All models were generated with the gradient boosting algorithm using samples from healthy adults only to classify Enterobacteriaceae colonization status.

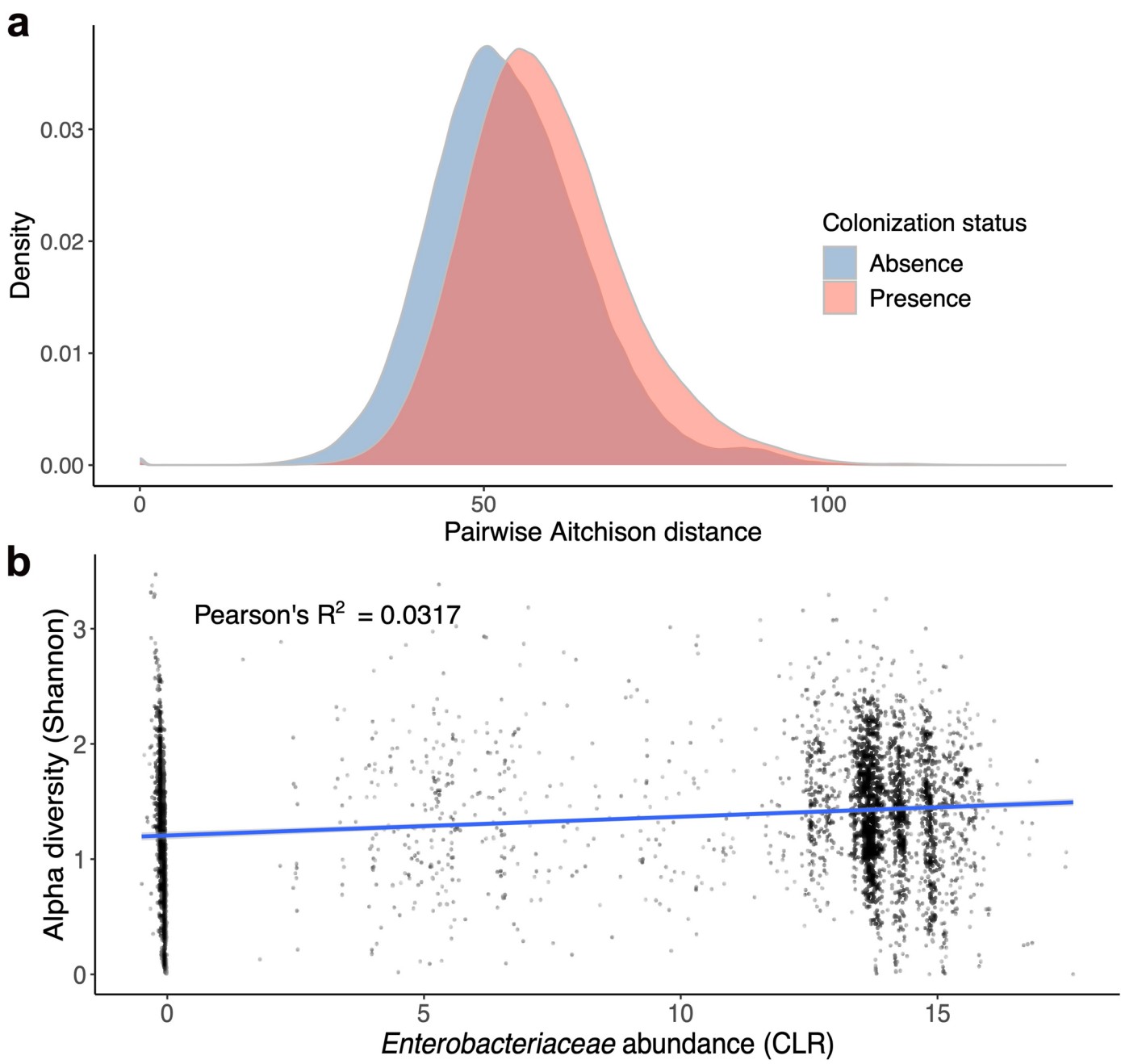

**Extended Data Fig. 4 | Microbiome diversity metrics based on Enterobacteriaceae colonization status and abundance. a**, Distribution of pairwise beta diversity estimates (Aitchison distance) between samples with or without Enterobacteriaceae. **b**, Two-sided Pearson correlation between Enterobacteriaceae abundance (transformed to centred log-ratio) and gut microbiome alpha diversity (Shannon index). Sample depths were rarefied to 500,000 reads.

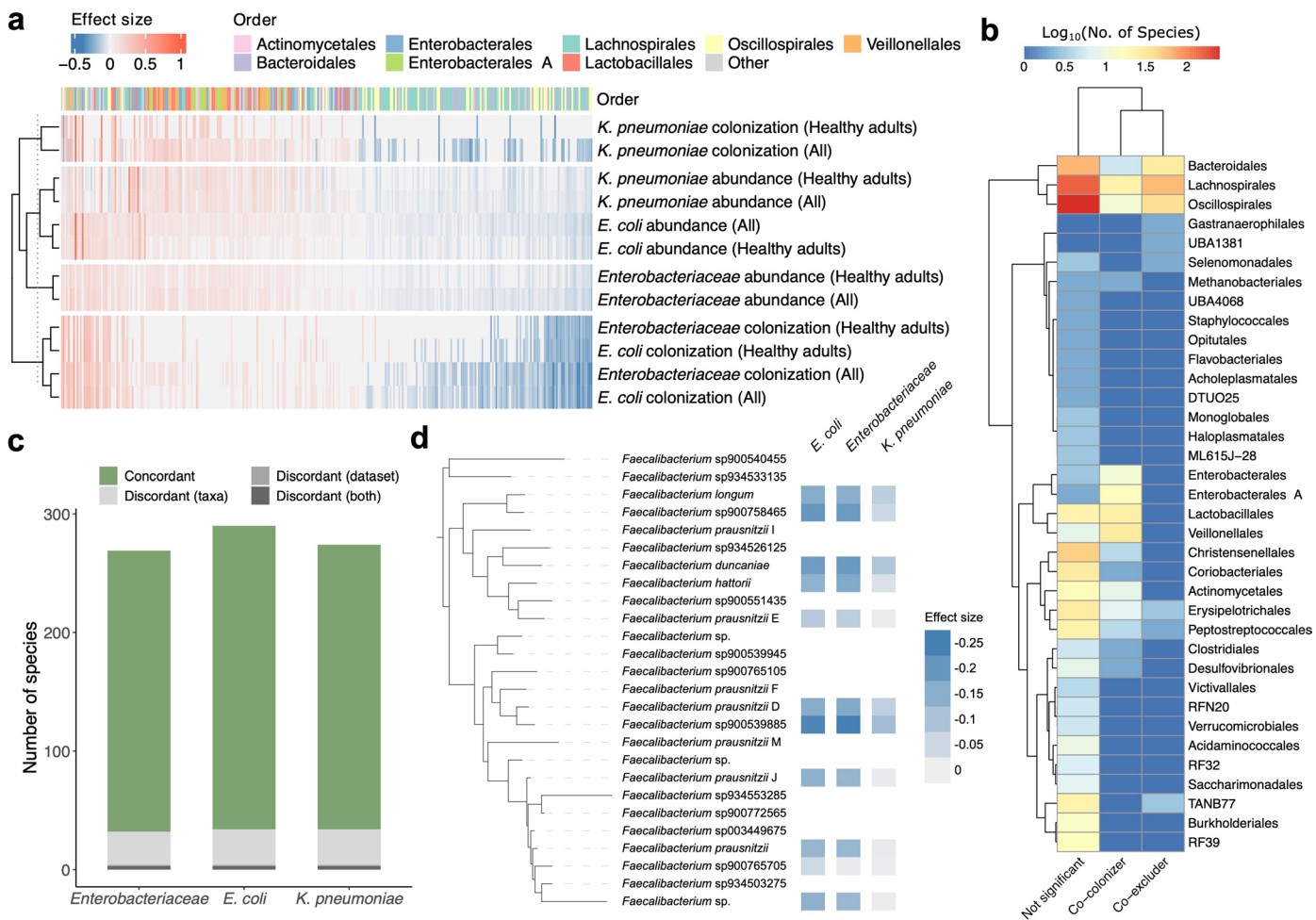

**Extended Data Fig. 5 | Candidate gut microbiome species associated with Enterobacteriaceae colonization and abundance. a**, Heatmap depicting all statistically significant microbiome species linked to Enterobacteriaceae, *E. coli* or *K. pneumoniae* colonization and/or abundance across the entire dataset or strictly among healthy adults. **b**, Number of species among the 1000 most prevalent detected that were classified as co-excluders, co-colonizers or not significant according to their order affiliation. **c**, Proportion of candidate species per taxon classified according to whether they were consistently associated to different taxa and/or across different datasets. **d**, Phylogenetic tree of representative genomes from all *Faecalibacterium* species detected in this study and their estimated association to Enterobacteriaceae, *E. coli* or *K. pneumoniae*. Species without a labeled effect size were not associated with any of the Enterobacteriaceae species tested.

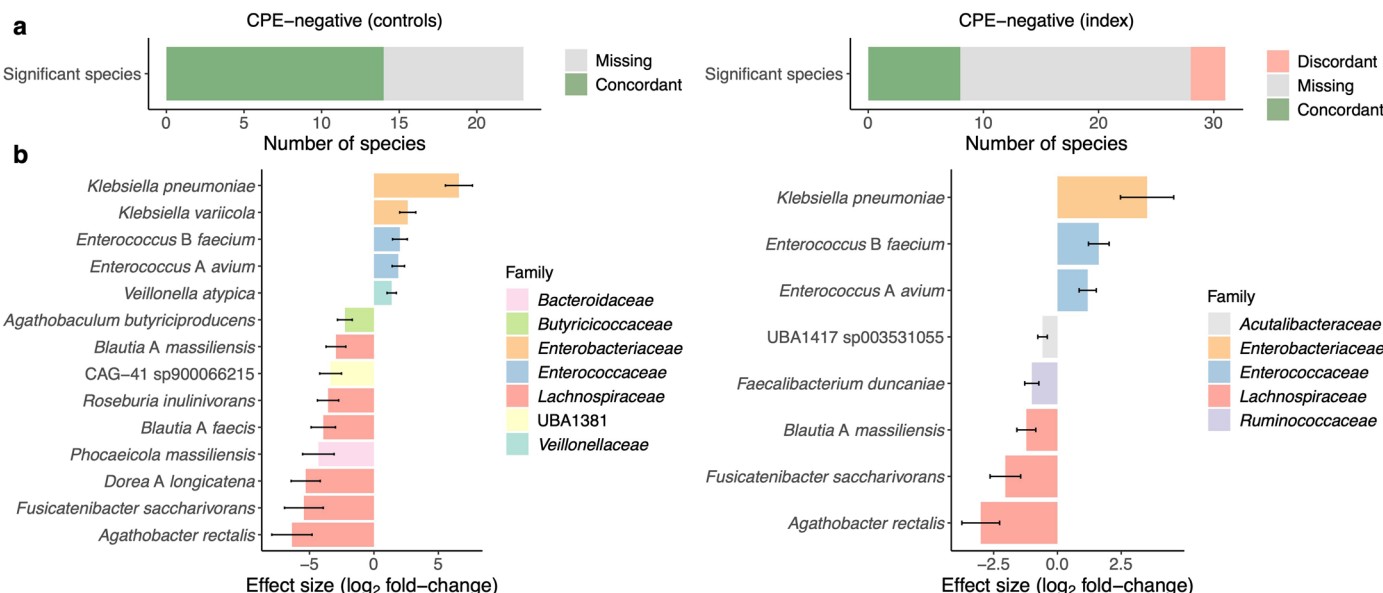

**Extended Data Fig. 6 | Co-excluders and co-colonizers of carbapenemase-producing Enterobacteriaceae. a**, Number of species differentially abundant between individuals colonized by carbapenemase-producing Enterobacteriaceae (CPE) compared to household negative controls (left) and compared to CPE-negative index subjects that were decolonized within the previous year (right). Species are coloured based on whether they were also found to be significantly different, and in the same direction, using the whole Enterobacteriaceae family (green), missing (grey) or significant but in opposite directions (red). **b**, Bar height represents the effect size derived from MaAsLin2 of species that were associated with both Enterobacteriaceae and CPE status using household controls (left) or using CPE-negative index subjects (right). Positive effect size denotes co-colonizers, while co-excluders are shown with a negative effect size. Error bars represent the standard error.

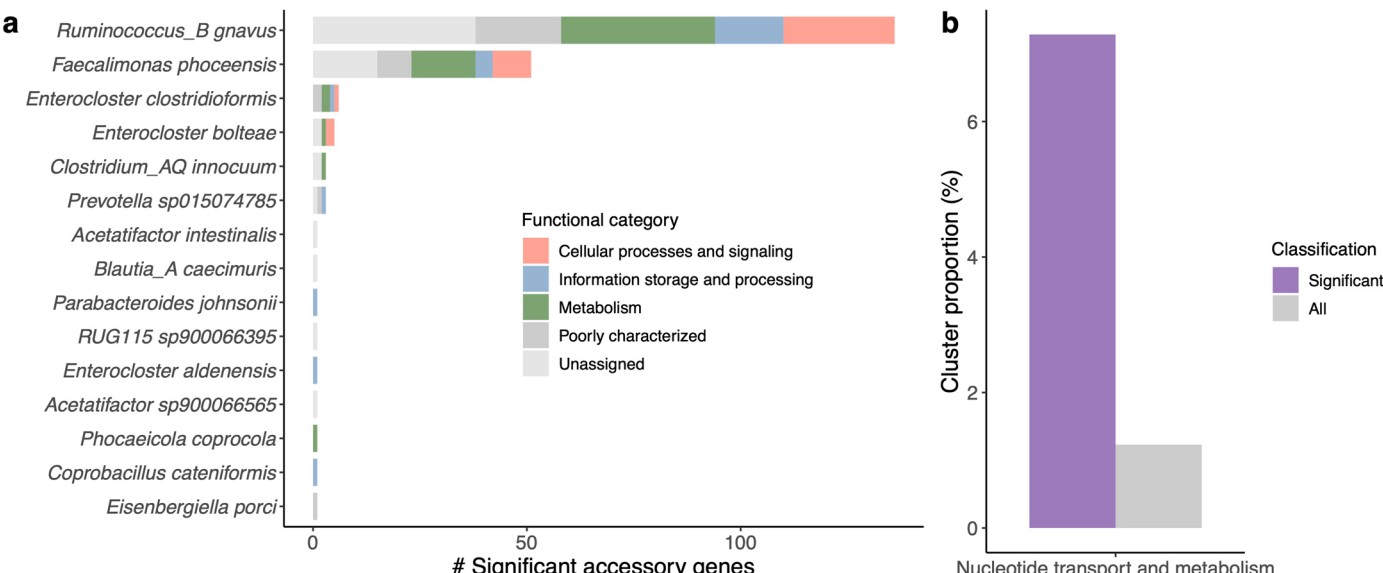

**Extended Data Fig. 7 | Accessory genes significantly linked to Enterobacteriaceae status. a**, Number and annotation of all accessory genes per species identified as significantly associated with Enterobacteriaceae colonization. Analysis was performed with 39 gut microbiome species that were identified as either co-colonizers or co-excluders of Enterobacteriaceae among healthy adults, but only 15 species contained significantly associated accessory genes. **b**, COG functional category significantly overrepresented (two-sided Fisher's exact test, adjusted $P = 7.93 \times 10^{-5}$) among the accessory genes associated with Enterobacteriaceae colonization.

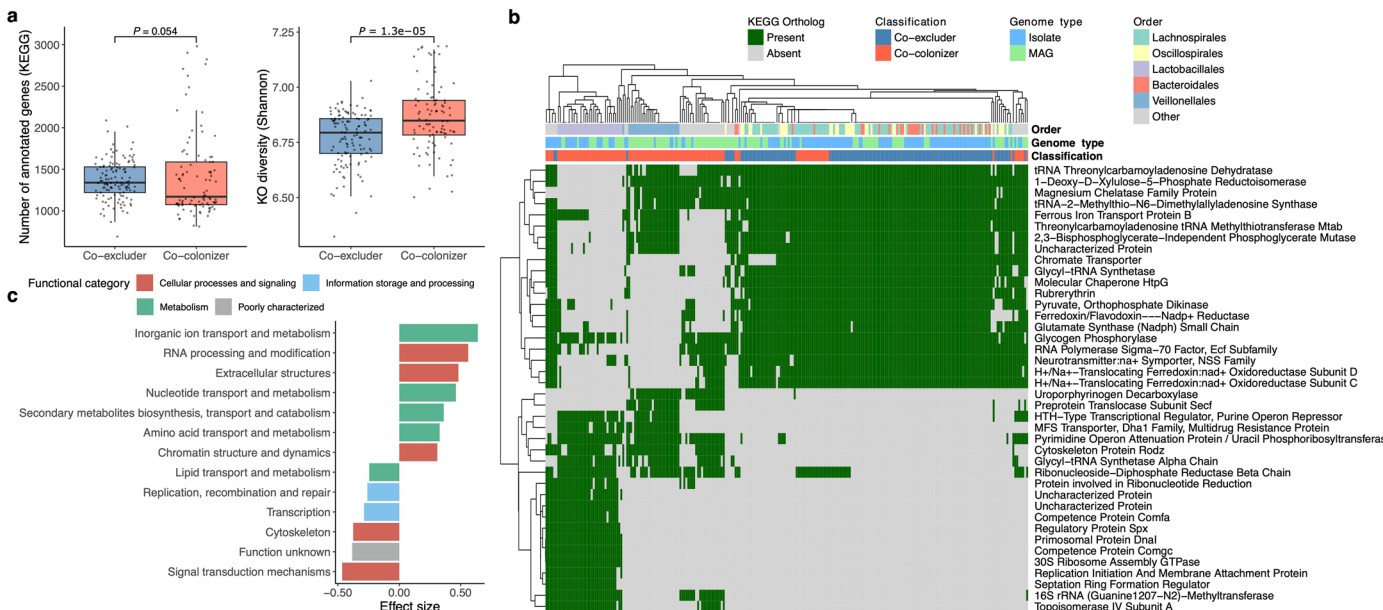

**Extended Data Fig. 8 | Functional diversity and candidate orthologs among co-excluders and co-colonizers. a**, Distribution of the number of annotated genes with KEGG (left) and Shannon diversity estimates (right) among co-excluders ($n = 122$) and co-colonizers ($n = 96$). Only genomes with >90% completeness were included. Box lengths represent the IQR of the data, the central line represents the median value, and the whiskers depict the lowest and highest values within 1.5 times the IQR of the first and third quartiles, respectively. *P* values were derived from a two-sided Wilcoxon rank-sum test. *P* values were derived from a Wilcoxon rank-sum test. **b**, Heatmap depicting the distribution of the top 20 KEGG Orthologs (KOs) associated with co-excluders or co-colonizers. Columns represent bacterial species coloured by their taxonomic affiliation, genome type and classification (co-colonizer or co-excluder). KOs are grouped using a complete linkage hierarchical clustering on the basis of their presence/absence patterns. **c**, COG functional categories significantly associated with co-colonizers (positive effect size) or co-excluders (negative effect size), only considering genomes belonging to the Bacillota phylum.

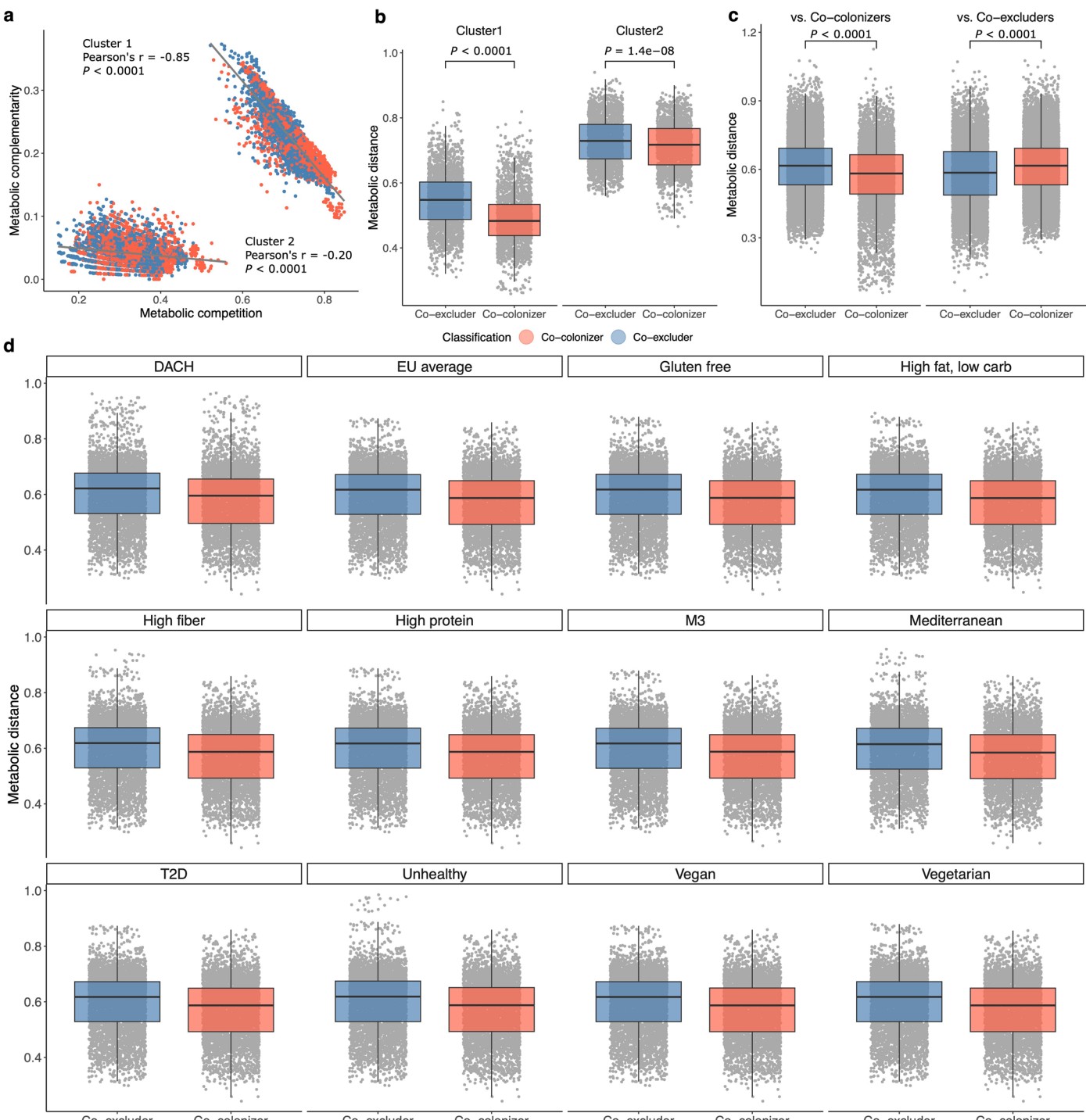

**Extended Data Fig. 9 | Metabolic indices estimated between gut microbiome species and Enterobacteriaceae. a**, Metabolic competition and complementary indices estimated with PhyloMint between co-excluders or co-colonizers and all Enterobacteriaceae species detected at >1% prevalence. **b**, Distribution of metabolic distance scores between co-colonizers ($n$ = 4292 comparisons) and co-excluders ($n$ = 4773 comparisons) in relation to Enterobacteriaceae. **c**, Comparison of metabolic distances within and between co-excluders and co-colonizers. Co-excluders vs. co-excluders: $n$ = 8256 comparisons; co-colonizers vs. co-colonizers: $n$ = 6670 comparisons; co-colonizers vs.

co-excluders: $n$ = 14,964 comparisons. **d**, Reproducibility of metabolic distance scores of co-colonizers ($n$ = 4292 comparisons) and co-excluders ($n$ = 4773 comparisons) compared to Enterobacteriaceae after simulating models with defined gut media (M1) supplemented with diets from the Virtual Metabolic Human database, or with the M3 rich growth media. All comparisons were statistically significant ($P$ < 0.0001). Box lengths represent the IQR of the data, the central line represents the median value, and the whiskers depict the lowest and highest values within 1.5 times the IQR of the first and third quartiles, respectively. $P$ values were derived from a two-sided Wilcoxon rank-sum test.

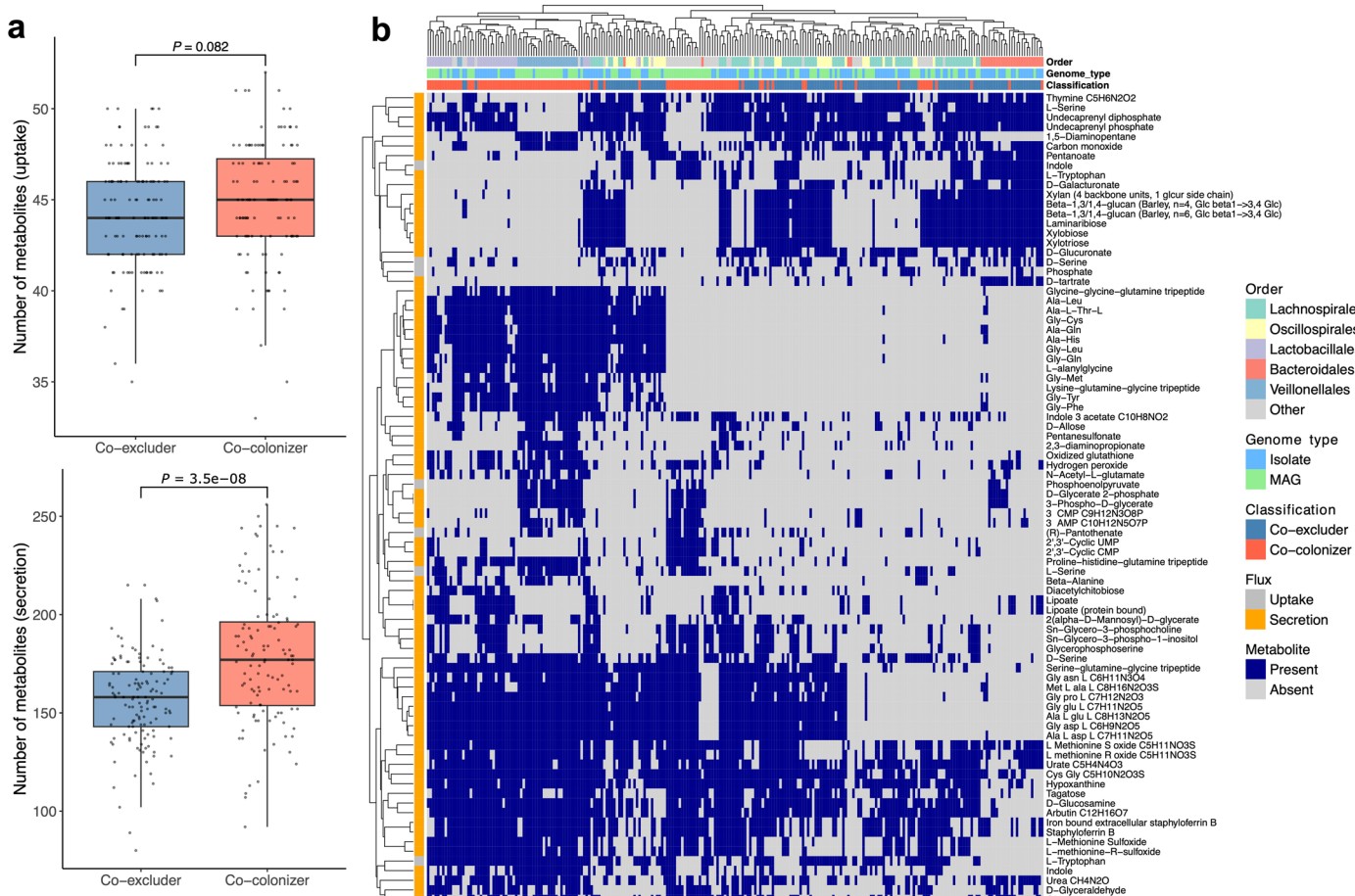

**Extended Data Fig. 10 | Distribution of predicted metabolites among co-excluders and co-colonizers. a**, Distribution of the number of metabolites predicted from uptake (top) or secretion (bottom) fluxes among co-excluders (*n* = 129) and co-colonizers (*n* = 116). *P* values were derived from a two-sided Wilcoxon rank-sum test. Box lengths represent the IQR of the data, the central line represents the median value, and the whiskers depict the lowest and highest values within 1.5 times the IQR of the first and third quartiles, respectively.

**b**, Metabolites significantly associated with either co-excluders or co-colonizers. Columns represent bacterial species coloured by their taxonomic affiliation, genome type and classification (co-colonizer or co-excluder). Metabolites are grouped using a complete linkage hierarchical clustering on the basis of their presence/absence patterns and coloured based on the type of metabolic flux (uptake or secretion).

# Reporting Summary

## Statistics

For all statistical analyses, confirm that the following items are present in the figure legend, table legend, main text, or Methods section.

| n/a | Confirmed | |
|---|---|---|
| ☐ | ☒ | The exact sample size (*n*) for each experimental group/condition, given as a discrete number and unit of measurement |
| ☒ | ☐ | A statement on whether measurements were taken from distinct samples or whether the same sample was measured repeatedly |
| ☐ | ☒ | The statistical test(s) used AND whether they are one- or two-sided<br>*Only common tests should be described solely by name; describe more complex techniques in the Methods section.* |
| ☐ | ☒ | A description of all covariates tested |
| ☐ | ☒ | A description of any assumptions or corrections, such as tests of normality and adjustment for multiple comparisons |
| ☐ | ☒ | A full description of the statistical parameters including central tendency (e.g. means) or other basic estimates (e.g. regression coefficient) AND variation (e.g. standard deviation) or associated estimates of uncertainty (e.g. confidence intervals) |
| ☐ | ☒ | For null hypothesis testing, the test statistic (e.g. *F*, *t*, *r*) with confidence intervals, effect sizes, degrees of freedom and *P* value noted<br>*Give P values as exact values whenever suitable.* |
| ☒ | ☐ | For Bayesian analysis, information on the choice of priors and Markov chain Monte Carlo settings |
| ☒ | ☐ | For hierarchical and complex designs, identification of the appropriate level for tests and full reporting of outcomes |
| ☐ | ☒ | Estimates of effect sizes (e.g. Cohen's *d*, Pearson's *r*), indicating how they were calculated |

*Our web collection on statistics for biologists contains articles on many of the points above.*

## Software and code

Policy information about availability of computer code

| Data collection | fastq-dl v2.0.4. Custom code: https://github.com/alexmsalmeida/metagen-fetch |
|---|---|
| Data analysis | TrimGalore v0.6.0; BWA MEM v0.7.16a-r1181; CheckM v1.0.11; GUNC v1.0.3; GTDB-Tk v2.3.2; Samtools v1.9; snakemake v7.32.3; ConQuR v1.2.0; Mikropml R package; vegan R package; FastSpar v1.0; ALDEx2 v1.32.0; MaAsLin2 v1.14.1; metaMLST v1.2.3; igraph R package; MEGAHIT v1.2.9; Prodigal v2.6.3; DIAMOND v2.1.8; Panaroo v1.3.3; FastTree v2.1.11; iTOL v6; Prokka v1.14.16; eggNOG-mapper v2.1.3; dbCAN2 v2.0.11; KOFam release 2021-11; gutSMASH v1.0; antiSMASH v6.0.1; CarveMe v1.5.2; PhyloMint v0.1.0; COBRApy v0.29. bowtie2 v2.5.3. Custom code: https://github.com/microfundiv-lab/EnteroEco |

For manuscripts utilizing custom algorithms or software that are central to the research but not yet described in published literature, software must be made available to editors and reviewers. We strongly encourage code deposition in a community repository (e.g. GitHub). See the Nature Portfolio guidelines for submitting code & software for further information.

# Data

Policy information about availability of data

All manuscripts must include a data availability statement. This statement should provide the following information, where applicable:

- Accession codes, unique identifiers, or web links for publicly available datasets
- A description of any restrictions on data availability
- For clinical datasets or third party data, please ensure that the statement adheres to our policy

All the metagenomic datasets used in this study are publicly available in the European Nucleotide Archive (see Supplementary Table 1 for all associated accession codes). The sequence databases used were retrieved from the Unified Human Gastrointestinal Genome (UHGG) catalog v1.0 and Unified Human Gastrointestinal Protein (UHGP-90) catalog v1.0. Abundance data estimated for the UHGG species and all metagenomic samples here included is available in: https://doi.org/10.6084/m9.figshare.27044341.v1. FASTA files of the BGCs detected with antiSMASH for all co-excluders and co-colonizers can be accessed in: https://doi.org/10.6084/m9.figshare.27044335.v1. Accession code of the human reference genome used for decontamination (GRCh38) is GCA_000001405.15.

# Research involving human participants, their data, or biological material

Policy information about studies with human participants or human data. See also policy information about sex, gender (identity/presentation), and sexual orientation and race, ethnicity and racism.

| | |
|---|---|
| Reporting on sex and gender | N/A |
| Reporting on race, ethnicity, or other socially relevant groupings | N/A |
| Population characteristics | N/A |
| Recruitment | N/A |
| Ethics oversight | N/A |

Note that full information on the approval of the study protocol must also be provided in the manuscript.

# Field-specific reporting

Please select the one below that is the best fit for your research. If you are not sure, read the appropriate sections before making your selection.

☒ Life sciences   ☐ Behavioural & social sciences   ☐ Ecological, evolutionary & environmental sciences

For a reference copy of the document with all sections, see nature.com/documents/nr-reporting-summary-flat.pdf

# Life sciences study design

All studies must disclose on these points even when the disclosure is negative.

| | |
|---|---|
| Sample size | We compiled 12,238 human gut metagenomic samples available in the European Nucleotide Archive (ENA) encompassing 65 different studies from 45 countries (Supplementary Table 1). No sample size calculation was performed. Our analyses revealed that with a subset of the data comprising 5,128 samples from healthy adults we were able to reproduce the results obtained with the full 12,238 dataset, suggesting this sample size is sufficient to identify consistent differences. |
| Data exclusions | Samples were selected based on the following criteria: 1) containing at least 500,000 paired-end metagenomic reads; 2) with available metadata on health state, age group and country of origin; 3) from individuals with no diagnosed acute infections; and 4) no reported antibiotic usage in the previous month. |
| Replication | Microbiome signatures linked to Enterobacteriaceae colonization and abundance were confirmed using a subset of samples for healthy adults and using the intersection of three bioinformatics tools (ALDEx2, MaAsLin2 and FastSpar). In addition, machine learning analyses were also confirmed by stratifying samples by continent and performing pairwise cross validation of samples from different continents. We further used study ERP133829 to assess which co-excluder and co-colonizer species were associated with carbapenemase-producing Enterobacteriaceae in particular. Bootstrapping for the FastSpar analysis was repeated 1000 times, whereas machine learning analyses were undertaken with a 5-fold cross-validation repeated 10 times. |
| Randomization | Samples were classified as Enterobacteriacea positive or negative based on the detection of any Enterobacteriaceae species using read mapping. Covariates ("Age group", "Continent","Health state", "Read depth" and "Study") were controlled using the generalized and mixed effects models implemented in ALDEx2 and MaAsLin2. In addition, results were confirmed with a subset of samples from healthy adults only. For the machine learning analyses, "Study" was used as a grouping factor to ensure samples from the same study were kept together in either the training or test dataset. |
| Blinding | Blinding was not relevant to this study, as samples were grouped and categorized based on the detection of Enterobacteriaceae species. |

# Reporting for specific materials, systems and methods

We require information from authors about some types of materials, experimental systems and methods used in many studies. Here, indicate whether each material, system or method listed is relevant to your study. If you are not sure if a list item applies to your research, read the appropriate section before selecting a response.

## Materials & experimental systems

| n/a | Involved in the study |
|-----|----------------------|
| ☒ ☐ | Antibodies |
| ☒ ☐ | Eukaryotic cell lines |
| ☒ ☐ | Palaeontology and archaeology |
| ☒ ☐ | Animals and other organisms |
| ☒ ☐ | Clinical data |
| ☒ ☐ | Dual use research of concern |
| ☒ ☐ | Plants |

## Methods

| n/a | Involved in the study |
|-----|----------------------|
| ☒ ☐ | ChIP-seq |
| ☒ ☐ | Flow cytometry |
| ☒ ☐ | MRI-based neuroimaging |

## Plants

| | |
|---|---|
| Seed stocks | N/A |
| Novel plant genotypes | N/A |
| Authentication | N/A |

