## [Peer Review File · Nature Microbiology]

Ecological dynamics of *Enterobacteriaceae* in the human gut microbiome across global populations

Corresponding Author: Dr Alexandre Almeida

Version 0:

Reviewer comments:

Reviewer #1

(Remarks to the Author)

The manuscript by Yin Q. et al. presents an interesting analysis of the Enterobacteriaceae family across more than 12 thousand human gut microbiome samples. The work is very well organized and written. The authors did extensive analyses to taxonomically and functionally profile co-occurring and co-excluding species with Enterobacteriaceae.

I have some comments, mainly about the wording and interpretation of the results, as the employed methods seem sound and reasonable.

Line 50, "the outgrowth and transmission of" the word transmission in this sentence should be further evaluated. Do we know that Enterobacteriaceae are shared across individuals? I'm not aware that species from the Enterobacteriaceae family are usually transmitted across individuals. Although an intriguing hypothesis, I think this statement should be backed up.

Lines 161-162, "These structural characteristics are indicative of more stable and resilient communities." I'm not sure the statement is supported. Do we have reference measures of more stable and resilient microbial communities? I might not be aware of other studies supporting these network parameters, in which case, the authors should cite them at the end of the sentence.

Lines 203-208, "E. faecalis has been previously shown to promote the growth and survival of E. coli in vitro and in vivo through the production of L-ornithine. However, the relationship between species of the Intestinibacter and Veillonella genera with Enterobacteriaceae has not been previously explored and may potentially underly new mechanisms associated with Enterobacteriaceae colonization success and outgrowth." Considering the functional profiling the authors performed, could they assess if the Intestinibacter and Veillonella species share similar functional characteristics as E. faecalis? Although it could likely be that this is driven by other and different biological interactions, it will be an interesting point to explore further.

Lines 224-227, "A further phylogenetic analysis identified a significantly stronger population structure associated with Enterobacteriaceae co-colonization among F. phoceensis strains (PERMANOVA, $R^2 = 0.74$, $P < 0.001$) compared to R. gnavus genomes (PERMANOVA, $R^2 = 0.02$, $P < 0.001$; Fig. 4)." it is not clear to me from the figure "where" the co-colonization information is, could you annotate the tree to show how the phylogenetic structure relate with the co-colonization? OR, I might have wrongly interpreted this part, which then would likely benefit from a re-writing to improve clarity.

Lines 230-231, "these results suggest there is a more specific, strain-level association between the diversity of Faecalimonas with Enterobacteriaceae co-colonization." this might be a too simple interpretation, based only on one aspect; could it be that the Faecalimonas share more functional similarities with Veillonella and Streptococcus species?

Lines 247-250, "As iron is an essential nutrient for many human pathogens, an enrichment of iron-utilizing genes among Enterobacteriaceae co-excluders suggests that colonization resistance by gut commensals could be conferred partly through limiting iron availability." how can you exclude that the absence of Enterobacteriaceae is allowing other species to utilize iron otherwise taken by Enterobacteriaceae? from the functional profiles could you assess if Enterobacteriaceae-absent individuals have, in general, more iron-utilizing genes from other species?

(Remarks on code availability)

Reviewer #2

(Remarks to the Author)

In the manuscript „The global ecological landscape of Enterobacteriaceae within the human gut microbiome , “ Yin and

colleagues perform a large-scale metagenomics analysis to analyze the prevalence and abundance of enterobacterial species in around 12000 samples and subsequently correlate the presence of other microbiota species. This enabled them to identify “co-colonizers” and “excluding species” and genetic elements associated with these different properties.

This study utilizes a wide range of complementary state-of-the-art bioinformatic approaches to define microbial functions associated with colonization levels with Enterobacteriaceae. This has potential translational value, guiding current efforts to promote the removal or suppression of harmful MDR Enterobacteriaceae from the gut.

I am aware of the inherent limitations of metagenomics-based studies, but I felt very conflicted when reading this exciting manuscript. This is clearly a very well-performed study and an impressive survey, but the focus on purely data analysis limits the conclusions that can be drawn. Moreover, most of the potentially underlying mechanisms for many of the associations are well studied, e.g. SCFA, SCFA-suppressing Enterobacteriaceae. Nevertheless, seeing those links, which were established in animal models and heavily disturbed microbiomes, being substantiated also in healthy individuals gives credit to the study and could trigger explorations of underexplored avenues such as quorum-sensing between different taxa in the gut.

Main comments:

Line 93ff: My first question concerns the study's detection limit for Enterobacteriaceae. Particularly for Enterobacteriaceae, the lack of detection using metagenomics has to be taken with a grain of salt, as their colonization levels can be low. Another point is the problem of relative abundance vs. absolute abundance. For example, a biomass-dense microbiome could be colonized to the same absolute colonization level as a biomass-poor microbiome. This point will be difficult to assess, but the authors could at least validate their detection limit for different Enterobacteriaceae by spiking known concentrations of Enterobacteriaceae into synthetic standards, thereby establishing robust detection limits.

Line 99ff: I am not sure whether I missed it, but is the complete data on detected species somewhere available? This would be valuable for other researchers.

Line 104ff: For the association of Enterobacteriaceae colonization levels to diseases: Did the authors control for the different geographical distribution across the studies? Were they always case-control studies, or were results compared across different regions, E.g., by studying RA patients from various areas of the globe?

Line 125: I am not an expert, but how well does MLST work for metagenomics data? What is the detection limit and the sensitivity/specificity of the analysis? I am a bit surprised that 76% of strains belong to STs not described before! What about global distribution?

Line 148ff: How did the ML training on all samples perform for specific regions? To support the conclusion, this comparison is required, in my opinion

Line 181: Were only prokaryotic species discovered to be associated to Enterobacteriaceae colonization levels? Or other commensal microbes as well? Also, would it be possible to visualize the colonizers/co-excluders compared to all detected species/1000 most prevalent species? This could help to visualize which taxonomic groups are underrepresented!?

Line 220: I have noted this in the next sections at several places: Which directions are the associations? Positive or negative? Also, since a geographical effect was observed in previous sections, how to control this in the strain analysis? Or has this been considered but not found to be relevant?

Line 235ff: Considering the possibility that the “co-colonizers” contain many bacterial studies that are more closely related to Enterobacteriaceae and also contain presumed pathogens triggers the question of whether the following section is biased by the annotation quality of the genomes. Can the authors exclude this? Please provide respective metrics.

Line 252ff: Similarly, to which degree do these analyses reflect taxonomic patterns? Is this more general rather than specific differences?

Line 305ff: Does the observation of an over-representation of indole uptake make sense as a co-excluder characteristic? I would have expected rather the opposite because it is well-known that many Enterobacteriaceae are also indole producers. A deeper discussion of this observation would be valuable.

Line 324ff: Are the results reported in a table to further explore? This is a novel aspect of the paper. Unfortunately, the authors seemingly didn't provide the resources for further exploration.

Line 347ff: The discussion is mainly well-balanced and provides the necessary context. The authors could also discuss the question of the cause or consequence of their observations, considering limitations such as relative vs. absolute abundances and the potential of Enterobacteriaceae to produce antimicrobial substances.

Line 385ff: Various competitions between Enterobacteriaceae beyond *K. michiganensis* have been described on a mechanical level. Examples are competitive interactions *E. coli* strains and other Enterobacteriaceae (Eberl et al., Cell Host Microbe, Sassone-Corsi M., Nature 2016) or also within the *Klebsiella* genus (Osbelt et al., Cell host Microbe 2021). The authors could include additional examples and discuss the differences between healthy individuals and disturbed microbiomes, such as in patients undergoing transplantation (Schluter et al., Cell Host Microbe 2023).

(Remarks on code availability)

Reviewer #3

(Remarks to the Author)

Summary: The authors present a large association study, looking at co-occurrences of commensal bacterial species, strains, and genes with bacteria within the Enterobacteriaceae family across ~12,000 human gut metagenomes from across the planet. Enterobacteriaceae contains several opportunistic pathogens, and looking at the features associated with the presence or absence of this group could be relevant to human health. Most of the study leverages regression, correlation, and machine learning analyses to look at associations between species, strains, gene functions, and Enterobacteriaceae. The authors also use genome-scale metabolic model reconstructions to compare metabolic competition/complementarity of gut bacterial taxa. The analyses are reasonably well-designed and the conclusions are supported by the results. My main critique is that many of these associations are somewhat known (e.g., Enterobacteriaceae occurrence/abundance is negatively associated with butyrate-producing taxa). Furthermore, it would be interesting to see some kind of experimental validation of the observed associations, to see if engraftment/exclusion/decolonization of Enterobacteriaceae could be controlled. I have some suggestions for improving the manuscript below.

Specific Comments:

- 1) I suggest removing the correlation network statistics (e.g., modularity, connectivity, etc.), as I don't see how they add to the analysis or provide any insight into the biology. The conclusions about 'stability' are not supported by the network or diversity analyses. You would need longitudinal data to measure stability.
- 2) In your machine learning analysis, you mention that you use cross-validation for both hyperparameter tuning and for testing. Does this mean that all your data sets were leveraged for hyperparameter tuning? If so, then your 'test' data is not independent of your model fitting. You should make sure that your test set(s) is(are) excluded from the hyperparameter tuning step.
- 3) Please add more detail in the main text about how the metabolic competition and complementarity metrics were calculated. How sensitive were these parameters to the medium constraint (M3)? Did you try a couple of different media that represent common human diets (e.g., see the diets available on the Virtual Metabolic Human site)?
- 4) Many of these associations make sense, and some are unsurprising. I suggest you focus a bit more in the discussion on which observations are truly novel/surprising.
- 5) Finally, I don't expect you to add experimental validation to this study (beyond the scope), but it would be nice to try to leverage existing data to validate some of your observations. For example, there are several fecal transplant studies that have stool metagenomic data from donors and recipients before and after treatment. Can the taxa/genes you identify here (or the ML models) predict which recipients will see engraftment of donor Enterobacteriaceae species/strains following treatment (and those who may exclude donor Enterobacteriaceae species/strains)?

(Remarks on code availability)

While I have not cloned the github repo and tried to rerun the code, I looked at the repository and it appears to be reasonably well-documented and thorough. The link is functional, and it appears all the analysis code is accounted for.

Decision Letter:

5th September 2024

Dear Dr Almeida,

Thank you for your patience while your manuscript "The global ecological landscape of *Enterobacteriaceae* within the human gut microbiome" was under peer-review at Nature Microbiology. It has now been seen by 3 referees, whose expertise and comments you will find at the end of this email. Although they find your work of some potential interest, they have raised a number of concerns that will need to be addressed before we can consider publication of the work in Nature Microbiology.

In particular, several referees note that the work presents confirmatory findings. As a result, we would encourage you to use your dataset to reveal new biological insights if possible. Additionally, several referees mention the lack of validation. Referee #3 suggests using existing datasets to do this. We would not require in vivo experimental validation but if in vitro work is possible, we would suggest including these experiments. Other technical concerns raised by each referee should be addressed.

Should further experimental data allow you to address these criticisms, we would be happy to look at a revised manuscript.

Please include a data availability statement as a separate section after Methods but before references, under the heading "Data Availability". This section should inform readers about the availability of the data used to support the conclusions of your study. This information includes accession codes to public repositories (data banks for protein, DNA or RNA sequences, microarray, proteomics data etc...), references to source data published alongside the paper, unique identifiers such as URLs to data repository entries, or data set DOIs, and any other statement about data availability. At a minimum, you should include the following statement: "The data that support the findings of this study are available from the corresponding author upon request", mentioning any restrictions on availability. If DOIs are provided, we also strongly encourage including these in the Reference list (authors, title, publisher (repository name), identifier, year). For more guidance on how to write this section please see: <http://www.nature.com/authors/policies/data/data-availability-statements-data-citations.pdf>

* If you have not done so already we suggest that you begin to revise your manuscript so that it conforms to our Resource format instructions at <http://www.nature.com/nmicrobiol/info/final-submission>. Refer also to any guidelines provided in this letter.

When submitting the revised version of your manuscript, please pay close attention to our [href="https://www.nature.com/nature-portfolio/editorial-policies/image-integrity">Digital Image Integrity Guidelines.](https://www.nature.com/nature-portfolio/editorial-policies/image-integrity) and to the following points below:

Link Redacted

Note: This url links to your confidential homepage and associated information about manuscripts you may have submitted or be reviewing for us. If you wish to forward this e-mail to co-authors, please delete this link to your homepage first.

Nature Microbiology is committed to improving transparency in authorship. As part of our efforts in this direction, we are now requesting that all authors identified as 'corresponding author' on published papers create and link their Open Researcher and Contributor Identifier (ORCID) with their account on the Manuscript Tracking System (MTS), prior to acceptance. This applies to primary research papers only. ORCID helps the scientific community achieve unambiguous attribution of all scholarly contributions. You can create and link your ORCID from the home page of the MTS by clicking on 'Modify my Springer Nature account'. For more information please visit www.springernature.com/orcid.

If you wish to submit a suitably revised manuscript we would hope to receive it within 6 months. If you cannot send it within this time, please let us know. We will be happy to consider your revision, even if a similar study has been accepted for publication at Nature Microbiology or published elsewhere (up to a maximum of 6 months).

Yours sincerely,

Reviewer Expertise:

- Referee #1: Gut microbiome, machine learning
- Referee #2: Microbiome, ecology, colonisation resistance
- Referee #3: Genome-scale Metabolic models, Metabolite flux simulations

Reviewer Comments:

Reviewer #1 (Remarks to the Author):

The manuscript by Yin Q. et al. presents an interesting analysis of the Enterobacteriaceae family across more than 12 thousand human gut microbiome samples. The work is very well organized and written. The authors did extensive analyses to taxonomically and functionally profile co-occurring and co-excluding species with Enterobacteriaceae.

I have some comments, mainly about the wording and interpretation of the results, as the employed methods seem sound and reasonable.

Line 50, "the outgrowth and transmission of" the word transmission in this sentence should be further evaluated. Do we know that Enterobacteriaceae are shared across individuals? I'm not aware that species from the Enterobacteriaceae family are usually transmitted across individuals. Although an intriguing hypothesis, I think this statement should be backed up.

Lines 161-162, "These structural characteristics are indicative of more stable and resilient communities." I'm not sure the statement is supported. Do we have reference measures of more stable and resilient microbial communities? I might not be aware of other studies supporting these network parameters, in which case, the authors should cite them at the end of the sentence.

Lines 203-208, "E. faecalis has been previously shown to promote the growth and survival of E. coli in vitro and in vivo through the production of L-ornithine. However, the relationship between species of the Intestinibacter and Veillonella genera with Enterobacteriaceae has not been previously explored and may potentially underly new mechanisms associated with Enterobacteriaceae colonization success and outgrowth." Considering the functional profiling the authors performed, could they assess if the Intestinibacter and Veillonella species share similar functional characteristics as E. faecalis? Although it could likely be that this is driven by other and different biological interactions, it will be an interesting point to explore further.

Lines 224-227, "A further phylogenetic analysis identified a significantly stronger population structure associated with Enterobacteriaceae co-colonization among F. phoceensis strains (PERMANOVA, $R^2 = 0.74$, $P < 0.001$) compared to R. gnavus genomes (PERMANOVA, $R^2 = 0.02$, $P < 0.001$; Fig. 4)." it is not clear to me from the figure "where" the co-colonization information is, could you annotate the tree to show how the phylogenetic structure relate with the co-colonization? OR, I might have wrongly interpreted this part, which then would likely benefit from a re-writing to improve clarity.

Lines 230-231, "these results suggest there is a more specific, strain-level association between the diversity of Faecalimonas with Enterobacteriaceae co-colonization." this might be a too simple interpretation, based only on one aspect; could it be that the Faecalimonas share more functional similarities with Veillonella and Streptococcus species?

Lines 247-250, "As iron is an essential nutrient for many human pathogens, an enrichment of iron-utilizing genes among Enterobacteriaceae co-excluders suggests that colonization resistance by gut commensals could be conferred partly through limiting iron availability." how can you exclude that the absence of Enterobacteriaceae is allowing other species to utilize iron otherwise taken by Enterobacteriaceae? from the functional profiles could you assess if Enterobacteriaceae-absent individuals have, in general, more iron-utilizing genes from other species?

Reviewer #2 (Remarks to the Author):

In the manuscript „The global ecological landscape of Enterobacteriaceae within the human gut microbiome“, Yin and colleagues perform a large-scale metagenomics analysis to analyze the prevalence and abundance of enterobacterial species in around 12000 samples and subsequently correlate the presence of other microbiota species. This enabled them to identify “co-colonizers” and “excluding species” and genetic elements associated with these different properties. This study utilizes a wide range of complementary state-of-the-art bioinformatic approaches to define microbial functions associated with colonization levels with Enterobacteriaceae. This has potential translational value, guiding current efforts to promote the removal or suppression of harmful MDR Enterobacteriaceae from the gut.

I am aware of the inherent limitations of metagenomics-based studies, but I felt very conflicted when reading this exciting manuscript. This is clearly a very well-performed study and an impressive survey, but the focus on purely data analysis limits the conclusions that can be drawn. Moreover, most of the potentially underlying mechanisms for many of the associations are well studied, e.g. SCFA, SCFA-suppressing Enterobacteriaceae. Nevertheless, seeing those links, which were established in animal models and heavily disturbed microbiomes, being substantiated also in healthy individuals gives credit to the study and could trigger explorations of underexplored avenues such as quorum-sensing between different taxa in the gut.

Main comments:

Line 93ff: My first question concerns the study's detection limit for Enterobacteriaceae. Particularly for Enterobacteriaceae, the lack of detection using metagenomics has to be taken with a grain of salt, as their colonization levels can be low. Another point is the problem of relative abundance vs. absolute abundance. For example, a biomass-dense microbiome could be colonized to the same absolute colonization level as a biomass-poor microbiome. This point will be difficult to assess, but the authors could at least validate their detection limit for different Enterobacteriaceae by spiking known concentrations of Enterobacteriaceae into synthetic standards, thereby establishing robust detection limits.

Line 99ff: I am not sure whether I missed it, but is the complete data on detected species somewhere available? This would be valuable for other researchers.

Line 104ff: For the association of Enterobacteriaceae colonization levels to diseases: Did the authors control for the different geographical distribution across the studies? Were they always case-control studies, or were results compared across different regions, E.g., by studying RA patients from various areas of the globe?

Line 125: I am not an expert, but how well does MLST work for metagenomics data? What is the detection limit and the sensitivity/specificity of the analysis? I am a bit surprised that 76% of strains belong to STs not described before! What about global distribution?

Line 148ff: How did the ML training on all samples perform for specific regions? To support the conclusion, this comparison is required, in my opinion

Line 181: Were only prokaryotic species discovered to be associated to Enterobacteriaceae colonization levels? Or other commensal microbes as well? Also, would it be possible to visualize the colonizers/co-excluders compared to all detected species/1000 most prevalent species? This could help to visualize which taxonomic groups are underrepresented!?

Line 220: I have noted this in the next sections at several places: Which directions are the associations? Positive or negative? Also, since a geographical effect was observed in previous sections, how to control this in the strain analysis? Or has this been considered but not found to be relevant?

Line 235ff: Considering the possibility that the "co-colonizers" contain many bacterial species that are more closely related to Enterobacteriaceae and also contain presumed pathogens triggers the question of whether the following section is biased by the annotation quality of the genomes. Can the authors exclude this? Please provide respective metrics.

Line 252ff: Similarly, to which degree do these analyses reflect taxonomic patterns? Is this more general rather than specific differences?

Line 305ff: Does the observation of an over-representation of indole uptake make sense as a co-excluder characteristic? I would have expected rather the opposite because it is well-known that many Enterobacteriaceae are also indole producers. A deeper discussion of this observation would be valuable.

Line 324ff: Are the results reported in a table to further explore? This is a novel aspect of the paper. Unfortunately, the authors seemingly didn't provide the resources for further exploration.

Line 347ff: The discussion is mainly well-balanced and provides the necessary context. The authors could also discuss the question of the cause or consequence of their observations, considering limitations such as relative vs. absolute abundances and the potential of Enterobacteriaceae to produce antimicrobial substances.

Line 385ff: Various competitions between Enterobacteriaceae beyond *K. michiganensis* have been described on a mechanical level. Examples are competitive interactions *E. coli* strains and other Enterobacteriaceae (Eberl et al., Cell Host Microbe, Sassone-Corsi M., Nature 2016) or also within the *Klebsiella* genus (Osbelt et al., Cell host Microbe 2021). The authors could include additional examples and discuss the differences between healthy individuals and disturbed microbiomes, such as in patients undergoing transplantation (Schluter et al., Cell Host Microbe 2023).

Reviewer #3 (Remarks to the Author):

Summary: The authors present a large association study, looking at co-occurrences of commensal bacterial species, strains, and genes with bacteria within the Enterobacteriaceae family across ~12,000 human gut metagenomes from across the planet. Enterobacteriaceae contains several opportunistic pathogens, and looking at the features associated with the presence or absence of this group could be relevant to human health. Most of the study leverages regression, correlation, and machine learning analyses to look at associations between species, strains, gene functions, and Enterobacteriaceae. The authors also use genome-scale metabolic model reconstructions to compare metabolic competition/complementarity of gut bacterial taxa. The analyses are reasonably well-designed and the conclusions are supported by the results. My main critique is that many of these associations are somewhat known (e.g., Enterobacteriaceae occurrence/abundance is negatively associated with butyrate-producing taxa). Furthermore, it would be interesting to see some kind of experimental validation of the observed associations, to see if engraftment/exclusion/decolonization of Enterobacteriaceae could be controlled. I have some suggestions for improving the manuscript below.

Specific Comments:

1) I suggest removing the correlation network statistics (e.g., modularity, connectivity, etc.), as I don't see how they add to the analysis or provide any insight into the biology. The conclusions about 'stability' are not supported by the network or diversity analyses. You would need longitudinal data to measure stability.

2) In your machine learning analysis, you mention that you use cross-validation for both hyperparameter tuning and for testing. Does this mean that all your data sets were leveraged for hyperparameter tuning? If so, then your 'test' data is not independent of your model fitting. You should make sure that your test set(s) is(are) excluded from the hyperparameter tuning step.

3) Please add more detail in the main text about how the metabolic competition and complementarity metrics were calculated. How sensitive were these parameters to the medium constraint (M3)? Did you try a couple of different media that represent common human diets (e.g., see the diets available on the Virtual Metabolic Human site)?

4) Many of these associations make sense, and some are unsurprising. I suggest you focus a bit more in the discussion on

which observations are truly novel/surprising.

5) Finally, I don't expect you to add experimental validation to this study (beyond the scope), but it would be nice to try to leverage existing data to validate some of your observations. For example, there are several fecal transplant studies that have stool metagenomic data from donors and recipients before and after treatment. Can the taxa/genes you identify here (or the ML models) predict which recipients will see engraftment of donor Enterobacteriaceae species/strains following treatment (and those who may exclude donor Enterobacteriaceae species/strains)?

Reviewer #3 (Remarks on code availability):

While I have not cloned the github repo and tried to rerun the code, I looked at the repository and it appears to be reasonably well-documented and thorough. The link is functional, and it appears all the analysis code is accounted for.

Version 1:

Reviewer comments:

Reviewer #1

(Remarks to the Author)

The revised version of the manuscript greatly improved, and all points raised were successfully taken into consideration, discussed and revised in the new version of the manuscript.

(Remarks on code availability)

Although I have not run the code myself, the repository appears to contain all the steps described by the authors with reasonable documentation.

Reviewer #2

(Remarks to the Author)

All my questions/comments have been addressed. I congratulate the authors on this comprehensive study.

(Remarks on code availability)

Reviewer #3

(Remarks to the Author)

The authors have done an excellent job addressing my prior comments. I have no further concerns.

(Remarks on code availability)

Decision Letter:

Our ref: NMICROBIOL-24072251A

18th October 2024

Dear Dr. Almeida,

Thank you for submitting your revised manuscript "The global ecological landscape of *Enterobacteriaceae* within the human gut microbiome" (NMICROBIOL-24072251A). It has now been seen by the original referees and their comments are below. The reviewers find that the paper has improved in revision, and therefore we'll be happy to publish it in Nature Microbiology, in principle, pending minor revisions to comply with our editorial and formatting guidelines.

Thank you again for your interest in Nature Microbiology. Please do not hesitate to contact me if you have any questions.

Happy Friday!!

Best,

Reviewer #1 (Remarks to the Author):

The revised version of the manuscript greatly improved, and all points raised were successfully taken into consideration, discussed and revised in the new version of the manuscript.

Reviewer #1 (Remarks on code availability):

Although I have not run the code myself, the repository appears to contain all the steps described by the authors with reasonable documentation.

Reviewer #2 (Remarks to the Author):

All my questions/comments have been addressed. I congratulate the authors on this comprehensive study.

Reviewer #3 (Remarks to the Author):

The authors have done an excellent job addressing my prior comments. I have no further concerns.

Version 2:

Decision Letter:

12th December 2024

Dear Alex,

I am pleased to accept your Analysis "Ecological dynamics of *Enterobacteriaceae* in the human gut microbiome across global populations" for publication in Nature Microbiology. Thank you for having chosen to submit your work to us and many congratulations.

You may wish to make your media relations office aware of your accepted publication, in case they consider it appropriate to organize some internal or external publicity. Once your paper has been scheduled you will receive an email confirming the publication details. This is normally 3-4 working days in advance of publication. If you need additional notice of the date and time of publication, please let the production team know when you receive the proof of your article to ensure there is sufficient time to coordinate. Further information on our embargo policies can be found here:

<https://www.nature.com/authors/policies/embargo.html>

Please note that *Nature Microbiology* is a Transformative Journal (TJ). Authors may publish their research with us through the traditional subscription access route or make their paper immediately open access through payment of an article-processing charge (APC). Authors will not be required to make a final decision about access to their article until it has been accepted. Find out more about Transformative

Journals

Authors may need to take specific actions to achieve [compliance](https://www.springernature.com/gp/open-research/funding/policy-compliance-faqs) with funder and institutional open access mandates. If your research is supported by a funder that requires immediate open access (e.g. according to [Plan S principles](https://www.springernature.com/gp/open-research/plan-s-compliance)) then you should select the gold OA route, and we will direct you to the compliant route where possible. For authors selecting the subscription publication route, the journal's standard licensing terms will need to be accepted, including [self-archiving policies](https://www.nature.com/nature-portfolio/editorial-policies/self-archiving-and-license-to-publish). Those licensing terms will supersede any other terms that the author or any third party may assert apply to any version of the manuscript.

With kind regards,

P.S. Click on the following link if you would like to recommend Nature Microbiology to your librarian
<http://www.nature.com/subscriptions/recommend.html#forms>

** Visit the Springer Nature Editorial and Publishing website at http://editorial-jobs.springernature.com?utm_source=ejp_NMicro_email&utm_medium=ejp_NMicro_email&utm_campaign=ejp_NMicro for more information about our career opportunities. If you have any questions please click [here](mailto:editorial.publishing.jobs@springernature.com).**

Reviewer #1 (Remarks to the Author):

The manuscript by Yin Q. et al. presents an interesting analysis of the Enterobacteriaceae family across more than 12 thousand human gut microbiome samples. The work is very well organized and written. The authors did extensive analyses to taxonomically and functionally profile co-occurring and co-excluding species with Enterobacteriaceae.

I have some comments, mainly about the wording and interpretation of the results, as the employed methods seem sound and reasonable.

We greatly appreciate the reviewer's positive feedback and have rephrased the wording and interpretation of the results throughout the manuscript as suggested below.

Line 50, "the outgrowth and transmission of" the word transmission in this sentence should be further evaluated. Do we know that Enterobacteriaceae are shared across individuals? I'm not aware that species from the Enterobacteriaceae family are usually transmitted across individuals. Although an intriguing hypothesis, I think this statement should be backed up.

We thank the reviewer for this suggestion. We have investigated this aspect further and have now cited a publication that looked at transmission rates of ESBL-*Enterobacteriaceae* within household and hospital settings (PMID: 22718774). We have added a sentence to further explain this point, as follows:

Lines 50-52: "Moreover, transmission rates of ESBL-producing *Enterobacteriaceae* in household settings — 23% for *E. coli* and 25% for *K. pneumoniae* — were found to surpass those in healthcare environments⁸."

Lines 161-162, "These structural characteristics are indicative of more stable and resilient communities." I'm not sure the statement is supported. Do we have reference measures of more stable and resilient microbial communities? I might not be aware of other studies supporting these network parameters, in which case, the authors should cite them at the end of the sentence.

Based on the suggestion by referee #3 we have now decided to exclude these results from the revised manuscript.

Lines 203-208, "E. faecalis has been previously shown to promote the growth and survival of E. coli in vitro and in vivo through the production of L-ornithine. However, the relationship between species of the Intestinibacter and Veillonella genera with Enterobacteriaceae has not been previously explored and may potentially underly new mechanisms associated with Enterobacteriaceae colonization success and outgrowth." Considering the functional profiling the authors performed, could they assess if the Intestinibacter and Veillonella species share similar functional characteristics as E. faecalis? Although it could likely be that this is driven by other and different biological interactions, it will be an interesting point to explore further.

We thank the reviewer for raising this interesting point. We have investigated further the functional similarities between co-excluders and co-colonizers and have now supplemented the metabolic modelling results with comparisons of co-excluders vs. co-colonizers; co-colonizers vs. co-colonizers; and co-excluders vs. co-excluders (see panel C in the new Extended Data Fig. 8 shown below).

Extended Data Figure 8. Metabolic indices estimated between gut microbiome species and *Enterobacteriaceae*. **a**, Metabolic competition and complementary indices estimated with PhyloMint between co-excluders or co-colonizers and all *Enterobacteriaceae* species detected at >1% prevalence. **b**, Distribution of metabolic distance scores between co-colonizers ($n=4292$ comparisons) and co-excluders ($n=4773$ comparisons) in relation to *Enterobacteriaceae*. **c**, Comparison of metabolic distances within and between co-excluders and co-colonizers. Co-excluders vs. co-excluders: $n = 8256$ comparisons; co-colonizers vs. co-colonizers: $n = 6670$ comparisons; co-colonizers vs. co-excluders: $n = 14,964$ comparisons. **d**, Reproducibility of metabolic distance scores compared to *Enterobacteriaceae* after simulating models with defined gut media (M1) supplemented with diets from the Virtual Metabolic Human database. All comparisons were statistically significant ($P < 0.0001$). Box lengths represent the IQR of the data, with whiskers depicting the lowest and highest values within 1.5 times the IQR of the first and third quartiles, respectively. P values were derived from a Wilcoxon rank-sum test.

We observed that species belonging to the same classification exhibit lower metabolic distances between them compared to other species (i.e., co-colonizers vs. co-colonizers are more similar

than co-colonizers vs. co-excluders). Even though we did not observe a notable pattern between *Intestinibacter/Veillonella/Enterococcus* in particular, these results still show that there are functional similarities between species that are positively or negatively associated with *Enterobacteriaceae*.

Beyond adding these results to Extended Data Fig. 8, we also discuss them in the revised manuscript as follows:

Lines 324-328: “Additionally, metabolic comparisons were made both within and between the groups of co-excluders and co-colonizers. These analyses revealed that within-group metabolic distances were smaller than between-group differences (Wilcoxon rank-sum test, $P < 0.0001$; Extended Data Fig. 8c), indicating shared niche preference.”

Lines 224-227, "A further phylogenetic analysis identified a significantly stronger population structure associated with *Enterobacteriaceae* co-colonization among *F. phoceensis* strains (PERMANOVA, $R^2 = 0.74$, $P < 0.001$) compared to *R. gnavus* genomes (PERMANOVA, $R^2 = 0.02$, $P < 0.001$; Fig. 4)." it is not clear to me from the figure "where" the co-colonization information is, could you annotate the tree to show how the phylogenetic structure relate with the co-colonization? OR, I might have wrongly interpreted this part, which then would likely benefit from a re-writing to improve clarity.

We apologize for the confusion. In this section, we investigated which strains within *F. phoceensis* and *R. gnavus* harboured the highest number of accessory genes associated with *Enterobacteriaceae* co-colonization. We found that the top 10% strains within *F. phoceensis* clustered very closely together, whereas for *R. gnavus* these were dispersed across the tree (red annotations in the outer layers of Fig. 4). We have clarified this section in the revised manuscript as follows:

Lines 243-248: “We further investigated the phylogenetic similarity of strains with the highest number of accessory genes associated with *Enterobacteriaceae* (top 10% strains; Fig. 4). This revealed a much stronger population structure associated with *Enterobacteriaceae* co-colonization among *F. phoceensis* strains (PERMANOVA, $R^2 = 0.74$, $P < 0.001$) compared to *R. gnavus* genomes (PERMANOVA, $R^2 = 0.02$, $P < 0.001$; Fig. 4).”

Lines 230-231, "these results suggest there is a more specific, strain-level association between the diversity of *Faecalimonas* with *Enterobacteriaceae* co-colonization." this might be a too simple interpretation, based only on one aspect; could it be that the *Faecalimonas* share more functional similarities with *Veillonella* and *Streptococcus* species?

We agree that our interpretation may be simplistic. Most of the accessory genes here discovered from *F. phoceensis* belong to unknown functions, thus it is challenging to make meaningful conclusions about the biological relevance of these particular strains. We have added this additional point in the revised manuscript:

Lines 252-254: “Future research avenues should focus on the unknown genes here identified to further understand the biological basis for the *F. phoceensis-Enterobacteriaceae* strain-specific co-colonization observed.”

Lines 247-250, "As iron is an essential nutrient for many human pathogens, an enrichment of iron-utilizing genes among *Enterobacteriaceae* co-excluders suggests that colonization

resistance by gut commensals could be conferred partly through limiting iron availability." how can you exclude that the absence of Enterobacteriaceae is allowing other species to utilize iron otherwise taken by Enterobacteriaceae? from the functional profiles could you assess if Enterobacteriaceae-absent individuals have, in general, more iron-utilizing genes from other species?

The reviewer makes an important point that discerning cause and consequence is challenging with our metagenomic data. We did identify that *Enterobacteriaceae*-absent individuals harbour more species with iron-utilizing genes. However, we agree that even still this does not conclusively show that co-excluders are directly inhibiting *Enterobacteriaceae* colonization through iron uptake. We have reworded this section to be more cautious in our interpretation:

Lines 275-277: "As iron is an essential nutrient for many human pathogens²⁸, an enrichment of iron-utilizing genes among co-excluders suggests there is competition for iron availability between *Enterobacteriaceae* and co-excluder species."

Reviewer #2 (Remarks to the Author):

In the manuscript „The global ecological landscape of Enterobacteriaceae within the human gut microbiome , “ Yin and colleagues perform a large-scale metagenomics analysis to analyze the prevalence and abundance of enterobacterial species in around 12000 samples and subsequently correlate the presence of other microbiota species. This enabled them to identify “co-colonizers” and “excluding species” and genetic elements associated with these different properties.

This study utilizes a wide range of complementary state-of-the-art bioinformatic approaches to define microbial functions associated with colonization levels with Enterobacteriaceae. This has potential translational value, guiding current efforts to promote the removal or suppression of harmful MDR Enterobacteriaceae from the gut.

I am aware of the inherent limitations of metagenomics-based studies, but I felt very conflicted when reading this exciting manuscript. This is clearly a very well-performed study and an impressive survey, but the focus on purely data analysis limits the conclusions that can be drawn. Moreover, most of the potentially underlying mechanisms for many of the associations are well studied, e.g. SCFA, SCFA-suppressing Enterobacteriaceae. Nevertheless, seeing those links, which were established in animal models and heavily disturbed microbiomes, being substantiated also in healthy individuals gives credit to the study and could trigger explorations of underexplored avenues such as quorum-sensing between different taxa in the gut.

We appreciate the reviewer’s valuable feedback. We agree that a key strength of our study is balancing the validation of previous findings in a larger cohort while offering novel insights with greater confidence. Below, we address the reviewer's specific comments.

Main comments:

Line 93ff: My first question concerns the study's detection limit for Enterobacteriaceae. Particularly for Enterobacteriaceae, the lack of detection using metagenomics has to be taken with a grain of salt, as their colonization levels can be low. Another point is the problem of relative abundance vs. absolute abundance. For example, a biomass-dense microbiome could be colonized to the same absolute colonization level as a biomass-poor microbiome. This point will be difficult to assess, but the authors could at least validate their detection limit for different Enterobacteriaceae by spiking known concentrations of Enterobacteriaceae into synthetic standards, thereby establishing robust detection limits.

We thank the reviewer for this great suggestion. Discerning relative from absolute quantification is a major challenge in the field, and we acknowledge this as one of our study’s limitations. However, as suggested by the referee, we have now quantified the detection limit of our metagenomic analysis approach using synthetic metagenomic communities.

Briefly, we simulated metagenomic communities with the 50 most prevalent species detected in our dataset at equal relative abundances. Thereafter, we spiked-in one *Enterobacteriaceae* species at various known relative abundances (from 0.0001% to 1% at 8 intervals) to infer the minimum detectable threshold. Each analysis was repeated for five *Enterobacteriaceae* species representing the top 5 most prevalent genera (*Escherichia coli*, *Klebsiella pneumoniae*, *Enterobacter hormaechei*, *Citrobacter freundii* and *Kluyvera ascorbata*) and three levels of sequencing depth based on the distribution observed for our dataset: low depth (first quartile) = 13 million reads; medium depth (median) = 31 million reads; high depth (third quartile) = 51

million reads. This represented a total of 120 synthetic communities. Results are shown in panel C of Extended Data Fig. 1 of the revised manuscript, illustrated below:

Extended Data Figure 1. Sample distribution and mapping quality control. **a**, Distribution of age groups, health states and continents of the 12,238 gut metagenomic samples. **b**, Comparison of taxonomic profiles and abundances of three mock community samples in relation to their expected proportions, estimated using the read mapping filtering parameters used in this study. **c**, Detection limit of our metagenomic approach evaluated with 120 synthetic metagenomics consisting of the top 50 most prevalent gut species and one *Enterobacteriaceae* species at a defined abundance across three levels of sequencing depth. Horizontal dashed line represents the minimum relative abundance at which the five *Enterobacteriaceae* species tested were detected. Abundance values are log-scaled. **d**, Pearson correlation between the number of samples with or without *Enterobacteriaceae* across the 65 studies.

Overall, at low sequencing depths we were able to detect all 5 *Enterobacteriaceae* species at a relative abundance as low as 0.01%. At medium depths, this decreased to 0.005% and at high depths to 0.003%. Therefore, we can conclude that for most of the metagenomic datasets here

included our detection limit is in the range of 0.003-0.01%. We have added this information in the Results and Methods section of the revised manuscript, as follows:

Lines 101-103: “Applying these parameters to synthetic metagenomes showed the detection limit of our metagenomic approach to be within a relative abundance of 0.003–0.01% (Extended Data Fig. 1c).”

Lines 675-687: “In addition, we generated synthetic metagenomic communities to establish the detection limit of our metagenomic analysis approach. Briefly, metagenomes containing the 50 most prevalent species detected in our dataset at equal relative abundances were simulated with ‘wgsim’ from the Samtools⁵⁶ software package. Thereafter, we spiked-in one *Enterobacteriaceae* species at various known relative abundances (from 0.0001% to 1% at 8 intervals) and processed the sample with the mapping approach described above. Each analysis was repeated for five *Enterobacteriaceae* species representing the top five most prevalent genera (*E. coli*, *K. pneumoniae*, *E. hormaechei*, *Citrobacter freundii* and *Kluyvera ascorbata*) and three levels of sequencing depth based on the distribution observed for our dataset: low depth (first quartile) = 13 million reads; medium depth (median) = 31 million reads; high depth (third quartile) = 51 million reads. This represented a total of 120 synthetic communities. The detection limit was defined as the minimum relative abundance at which all five species were detected.”

Line 99ff: I am not sure whether I missed it, but is the complete data on detected species somewhere available? This would be valuable for other researchers.

Apologies for this omission. We have now uploaded the abundance data of all UHGG species across all metagenomic samples to Figshare and updated the ‘Data availability’ statement accordingly:

Lines 843-844: “Abundance data estimated for the UHGG species and all metagenomic samples here included is available in: <https://doi.org/10.6084/m9.figshare.27044341.v1>.”

Line 104ff: For the association of *Enterobacteriaceae* colonization levels to diseases: Did the authors control for the different geographical distribution across the studies? Were they always case-control studies, or were results compared across different regions, E.g., by studying RA patients from various areas of the globe?

We apologize for the confusion, but we would like to clarify that we did not explicitly perform an analysis associating *Enterobacteriaceae* with specific diseases. The numbers reported in this section are simply the total prevalence (number of samples that *Enterobacteriaceae* was detected) for different metadata categories. We only used this information to rank the species based on their frequency. However, when identifying co-excluding and co-excluder species, we did include the specific “disease state” as a co-variate in the linear models, as well as their continent of origin and age group (see further details in lines 722-731). In most cases, the diseases here included came from case-control studies where both healthy and diseased individuals were represented.

Line 125: I am not an expert, but how well does MLST work for metagenomics data? What is the detection limit and the sensitivity/specificity of the analysis? I am a bit surprised that 76% of strains belong to STs not described before! What about global distribution?

The pipeline we applied (metaMLST) was developed and validated in a prior study (Zolfo et al. 2017: PMID: 27651451). In the original paper, the authors report >98.5% accuracy at coverages as low as 1×.

Interestingly, based on the reviewer’s suggestion we investigated the geographical distribution of the STs and observed an overrepresentation of samples from Africa containing undescribed STs. We speculate this might be one of the reasons why many of these strains remain unclassified. We have added this information in the revised manuscript, as well as in panel D of Figure 2 (below).

Lines 133-134. “These novel lineages were found to be overrepresented particularly among samples from Africa (Fig. 2d).”

Figure 2. Distribution and diversity of the most prevalent *Enterobacteriaceae* species. **a**, Prevalence and median abundance of representative species from the five most prevalent *Enterobacteriaceae* genera across different age groups, continents and health states. **b**, Upset plot showing *Enterobacteriaceae* co-colonization patterns. Vertical bars represent the proportion of samples by continent harbouring the species highlighted in the lower panel. Numbers below the bars indicate sample size. Horizontal bars in the lower panel show the total number of samples each species was detected. **c**, Minimum spanning tree of the *E. coli* sequence types (STs) detected across 5,128 human gut metagenomes from healthy adults. The most prevalent STs are labelled next to their respective nodes (ST100024 and ST100083 represent novel STs). **d**, Geographical distribution of samples containing known or novel STs.

Line 148ff: How did the ML training on all samples perform for specific regions? To support the conclusion, this comparison is required, in my opinion

We thank the reviewer for this suggestion. We have added a new analysis, where the machine learning models were evaluated on a per continent basis. We observed that the ML models still performed well, even after stratifying by region (AUROC >0.7 for all comparisons). We have added these new data in the revised manuscript and in panel B of Extended Data Fig. 2.

Lines 154-156: “Focusing on samples from healthy adults, models tested on a per continent basis had overall good performance (AUROC >0.7 for all continents tested, Extended Data Fig. 2b), with the highest performance observed for samples from Africa.”

Extended Data Figure 2. Machine learning models to classify *Enterobacteriaceae* colonization status. **a**, Area Under the ROC Curve (AUROC) performance results of different machine learning methods, datasets and outcome variables (taxa) relating the gut microbiome composition with *Enterobacteriaceae* colonization status ($n=10$ independent seeds per analysis). Box lengths represent the IQR of the data, with whiskers depicting the lowest and highest values within 1.5 times the IQR of the first and third quartiles, respectively. **b**, ROC curve of the machine learning results linking the gut microbiome composition with *Enterobacteriaceae* status. AUROC values represent the median of gradient boosting models across 10 independent seeds, stratified by continent and only considering samples from healthy adults **c**, All-against-all performance results comparing models trained and tested using microbiome samples across different continents. All models were generated with the gradient boosting algorithm using samples from healthy adults only to classify *Enterobacteriaceae* colonization status.

Line 181: Were only prokaryotic species discovered to be associated to *Enterobacteriaceae* colonization levels? Or other commensal microbes as well? Also, would it be possible to visualize the colonizers/co-excluders compared to all detected species/1000 most prevalent species? This could help to visualize which taxonomic groups are underrepresented!?

To answer the first point, our study specifically investigated the relationship between *Enterobacteriaceae* and other prokaryotic gut species. We did not screen for eukaryotic organisms or viruses, as current annotation tools and databases (such as the UHGG) have been developed primarily for prokaryotic identification and would likely underestimate the contribution of other taxa and their functions to *Enterobacteriaceae* colonization dynamics.

We thank the reviewer for the great suggestion of visualizing the most prevalent species to identify underrepresented groups. We have added panel C to Fig. 3 (depicted below) showing the taxonomic distribution of the 1000 most prevalent species according to their classification (co-excluder, co-colonizer or not significant).

Figure 3. Gut microbiome composition is associated with *Enterobacteriaceae* colonization and abundance. **a**, Receiver operating characteristic (ROC) curve of the machine learning results linking the gut microbiome composition with *Enterobacteriaceae*, *E. coli* or *K. pneumoniae* colonization status. Area Under the ROC Curve (AUROC) values were obtained with gradient boosting applied to the 12,238 human gut metagenomes. **b**, Phylogenetic tree of the 306 bacterial species associated with *Enterobacteriaceae* colonization and abundance. Clades are coloured according to their affiliated order. Red and blue colours in the outer layer indicate the number of analyses (out of 12) in which each species was classified as a co-excluder (negative) or co-colonizer (positive). A maximum score of 12 denotes that species was found to be consistently associated with *Enterobacteriaceae*, *E. coli* and *K. pneumoniae* co-colonization across all 12,238 datasets, as well as the subset of healthy adults. **c**, Number of species among the 1000 most prevalent detected that were classified as co-excluders, co-colonizers or not significant according to their order affiliation. **d**, Top 10 gut microbiome species classified as co-excluders (negative effect size) or co-colonizers (positive effect size) coloured by their family affiliation.

We discovered there were 17 bacterial orders with no taxa classified as significantly associated with *Enterobacteriaceae*, with two in particular (RF39 and Burkholderiales) represented by

>10 species each among the 1000 most prevalent species. We have added these new results in the manuscript as follows:

Lines 195-197. “Analysis of the 1000 most prevalent species revealed that 17 bacterial orders contained neither co-excluders nor co-colonizers (Fig. 3c), even though two taxa in particular (RF39 and Burkholderiales) were represented by >10 species each.”

Line 220: I have noted this in the next sections at several places: Which directions are the associations? Positive or negative? Also, since a geographical effect was observed in previous sections, how to control this in the strain analysis? Or has this been considered but not found to be relevant?

Of the 213 genes, 207 were positively- and 6 negatively-associated with *Enterobacteriaceae*. We have now clarified this in the text. Regarding the geographical effect, we did not explicitly control for geography for this analysis (only “Study” was used as a covariate to account for batch effects). However, visualization of the *F. phoceensis* species phylogeny (Fig. 4, left) shows that the cluster of strains associated with *Enterobacteriaceae* co-colonization (annotated in red in the outer layer) are found across all continents where the *F. phoceensis* genomes were recovered (Asia, Europe and North America). This suggests that geographical location was not a confounding factor in this analysis.

Line 235ff: Considering the possibility that the “co-colonizers” contain many bacterial studies that are more closely related to Enterobacteriaceae and also contain presumed pathogens triggers the question of whether the following section is biased by the annotation quality of the genomes. Can the authors exclude this? Please provide respective metrics.

The reviewer raises an interesting point. To verify this, we have added two additional analyses to panel A of Extended Data Fig. 7 shown below.

Extended Data Figure 7. Functional diversity and candidate orthologs among co-excluders and co-colonizers. **a**, Distribution of the number of annotated genes with KEGG (left) and Shannon diversity estimates (right) among co-excluders ($n=122$) and co-colonizers ($n=96$). Only genomes with >90% completeness were included. Box lengths represent the IQR of the data, with whiskers depicting the lowest and highest values within 1.5 times the IQR of the first and third quartiles, respectively. P values were derived from a Wilcoxon rank-sum test. P values were derived from a

Wilcoxon rank-sum test. **b**, Heatmap depicting the distribution of the top 20 KEGG Orthologs (KOs) associated with co-excluders or co-colonizers. Columns represent bacterial species coloured by their taxonomic affiliation, genome type and classification (co-colonizer or co-excluder). KOs are grouped using a complete linkage hierarchical clustering on the basis of their presence/absence patterns. **c**, COG functional categories significantly associated with co-colonizers (positive effect size) or co-excluders (negative effect size), only considering genomes belonging to the Bacillota phylum.

Briefly, to account for the effect of both annotation coverage and genome quality, we compared the distribution of the number of annotated genes with KEGG between co-excluders and co-colonizers and further repeated the KO diversity analyses by only considering genomes with >90% completeness. These results showed that there was no significant difference between the number of annotated genes between co-excluders and co-colonizers, while the KO diversity differences were even more pronounced when filtering for higher quality genomes. We believe these results reinforce the fact that co-colonizers are more functionally diverse, regardless of differences in annotation coverage and genome quality. We have added these new data in the revised manuscript:

Lines 261-265: “Based on the annotation of protein-coding sequences with KEGG²⁶, we found that co-colonizers exhibited a greater functional diversity compared to co-excluders (Wilcoxon rank-sum test, $P = 0.0049$; Fig. 5a), which was found to be independent of annotation coverage and genome quality differences (Extended Data Fig. 7a).”

Line 252ff: Similarly, to which degree do these analyses reflect taxonomic patterns? Is this more general rather than specific differences?

Our functional analyses compared general differences between co-excluders and co-colonizers, which are likely to reflect some taxonomic patterns given the overrepresentation of specific taxa with either co-exclusion or co-colonization. However, we have further confirmed the COG results by only considering genomes belonging to the Bacillota phylum (the most represented phylum with both co-excluders and co-colonizers; see Extended Data Fig. 7c included in the previous comment). This confirmed that 12 out of 15 functional categories that were significant when analysing all candidate species also replicate when considering those from Bacillota alone. We have added this information in the revised manuscript.

Lines 288-290: “These results were consistent even when only considering species from the Bacillota phylum, which contains the highest number of both co-excluders and co-colonizers (Extended Data Fig. 7c).”

Line 305ff: Does the observation of an over-representation of indole uptake make sense as a co-excluder characteristic? I would have expected rather the opposite because it is well-known that many Enterobacteriaceae are also indole producer. A deeper discussion of this observation would be valuable.

We thank the reviewer for raising this point. Indole has been described as an important signalling molecule with roles in spore formation, drug resistance, biofilm formation and virulence in indole-producing bacteria. We therefore speculate that a higher uptake of indole among co-excluders could disrupt intercellular communication and impair certain physiological processes of *Enterobacteriaceae*. We have expanded the discussion of this result in the revised manuscript:

Lines 341-349. “With regards to estimated uptake fluxes, L-serine and indole were the most significant metabolites enriched in co-colonizers and co-excluders, respectively. Interestingly, dietary L-serine has been previously described to provide a competitive fitness advantage to *Enterobacteriaceae* under inflammatory conditions³⁵, while indole has been shown to alleviate intestinal inflammation through modulation of the gut microbiome composition³⁶. Indole is also recognized as a signalling molecule among indole-producing bacteria, such as *E. coli*³⁷. Therefore, a higher uptake of indole among co-excluders could impair intercellular signal communication of *Enterobacteriaceae*.”

Line 324ff: Are the results reported in a table to further explore? This is a novel aspect of the paper. Unfortunately, the authors seemingly didn't provide the resources for further exploration.

We apologize for this omission. We have now made the sequences of all biosynthetic gene clusters detected in our dataset available in Figsare (<https://doi.org/10.6084/m9.figshare.27044335.v1>) and updated the ‘Data availability’ statement accordingly. Furthermore, we have also added a new table (Supplementary Table 4) with the sequence alignment results that were used to build the network in Fig. 6b.

Line 347ff: The discussion is mainly well-balanced and provides the necessary context. The authors could also discuss the question of the cause or consequence of their observations, considering limitations such as relative vs. absolute abundances and the potential of *Enterobacteriaceae* to produce antimicrobial substances.

We have added these additional points in the revised manuscript.

Lines 399-406. “Despite representing the largest metagenomic investigation of *Enterobacteriaceae*-microbiome dynamics thus far, our study has some inherent limitations. Due to the compositional nature of sequencing data, we were unable to differentiate between absolute and relative abundance estimates. Metagenomic samples may show varying relative abundances of *Enterobacteriaceae*, yet reflect similar absolute colonization levels. Additionally, it remains unclear whether co-excluders are a cause or consequence of reduced *Enterobacteriaceae* levels, as species such as *E. coli* could potentially inhibit co-excluder growth through mechanisms such as antimicrobial production.”

Line 385ff: Various competitions between *Enterobacteriaceae* beyond *K. michiganensis* have been described on a mechanical level. Examples are competitive interactions *E. coli* strains and other *Enterobacteriaceae* (Eberl et al., Cell Host Microbe, Sassone-Corsi M., Nature 2016) or also within the *Klebsiella* genus (Osbelt et al., Cell host Microbe 2021). The authors could include additional examples and discuss the differences between healthy individuals and disturbed microbiomes, such as in patients undergoing transplantation (Schluter et al., Cell Host Microbe 2023).

We thank the reviewer for these suggestions and have revised the discussion accordingly.

Lines 414-425. “Prior studies in animal models⁴⁵⁻⁴⁹ and perturbed microbiomes⁵⁰ have described competitive interactions among *Enterobacteriaceae* species, including between *E. coli* strains and other *Enterobacteriaceae* species or between members of the *Klebsiella* genus specifically. However, in our study we detected a significant co-occurrence of species that are predicted to metabolically compete, including those within the *Enterobacteriaceae* family.

Therefore, our results suggest that habitat filtering has a stronger effect on *Enterobacteriaceae* colonization success in the human gut compared to that observed in animal models, where the impact of direct interspecies competition may be more pronounced. In addition, most of our findings were inferred from healthy populations, which exhibit substantial microbiome differences compared to disturbed microbiomes such as those undergoing transplantation or antibiotic treatment. Altogether, these data indicate that gut environmental conditions, for instance mediated by diet, play a key role in the risk of *Enterobacteriaceae* outgrowth.”

Reviewer #3 (Remarks to the Author):

Summary: The authors present a large association study, looking at co-occurrences of commensal bacterial species, strains, and genes with bacteria within the Enterobacteriaceae family across ~12,000 human gut metagenomes from across the planet. Enterobacteriaceae contains several opportunistic pathogens, and looking at the features associated with the presence or absence of this group could be relevant to human health. Most of the study leverages regression, correlation, and machine learning analyses to look at associations between species, strains, gene functions, and Enterobacteriaceae. The authors also use genome-scale metabolic model reconstructions to compare metabolic competition/complementarity of gut bacterial taxa. The analyses are reasonably well-designed and the conclusions are supported by the results. My main critique is that many of these associations are somewhat known (e.g., Enterobacteriaceae occurrence/abundance is negatively associated with butyrate-producing taxa). Furthermore, it would be interesting to see some kind of experimental validation of the observed associations, to see if engraftment/exclusion/decolonization of Enterobacteriaceae could be controlled. I have some suggestions for improving the manuscript below.

We thank the reviewer for their helpful feedback and have revised the manuscript based on their suggestions below.

Specific Comments:

1) I suggest removing the correlation network statistics (e.g., modularity, connectivity, etc.), as I don't see how they add to the analysis or provide any insight into the biology. The conclusions about 'stability' are not supported by the network or diversity analyses. You would need longitudinal data to measure stability.

We agree with the reviewer and have now excluded this analysis from the revised manuscript.

2) In your machine learning analysis, you mention that you use cross-validation for both hyperparameter tuning and for testing. Does this mean that all your data sets were leveraged for hyperparameter tuning? If so, then your 'test' data is not independent of your model fitting. You should make sure that your test set(s) is(are) excluded from the hyperparameter tuning step.

We apologize for the confusion. We can confirm that hyperparameter tuning was performed independently on the training dataset without the test data. Our pipeline used the mikropml R package for model training and testing. You can find below a schematic showing how the models are trained and tested based on the methodology described in the original publication of Topçuoğlu et al. mBio 2020, PMID: 32518182.

“Figure 1. Machine learning pipeline. We split the data to create a training (80%) and held-out test set (20%). The splits were stratified to maintain the overall class distribution. We performed five-fold cross-validation on the training data to select the best hyperparameter setting and then used these hyperparameters to train the models. The model was evaluated on the held-out data set. Abbreviations: cvAUC, cross-validation area under the receiver operating characteristic curve.” (Topçuoğlu et al. mBio 2020)

To make it clearer, we have revised the methods section as follows:

Lines 704-707: “Model training and hyperparameter tuning was performed on 80% of the data using a 5-fold cross-validation, while the other 20% were used for testing with the best hyperparameter setting. The whole procedure was then repeated 10 times with independent seeds.”

3) Please add more detail in the main text about how the metabolic competition and complementarity metrics were calculated. How sensitive were these parameters to the medium constraint (M3)? Did you try a couple of different media that represent common human diets (e.g., see the diets available on the Virtual Metabolic Human site)?

We greatly appreciate the reviewer’s suggestion. We have added more detail in the main text about the metabolic competition and complementarity scores.

Lines 313-315. “Metabolic competition was calculated based on the overlap between two given metabolic networks, while metabolic complementarity measured the potential of one species to utilize the metabolic output of another.”

We have also further validated the metabolic indices by simulating the models with different diets from the Virtual Metabolic Human database. Results are presented in panel D of Extended Data Fig. 8 shown below. These confirmed that the metabolic distances inferred are consistent across different media and diet compositions. We have added this new information in the revised manuscript:

Lines 328-331. “Importantly, results were consistent even when simulating metabolic models with different growth media and diets (Extended Data Fig. 8d).”

Extended Data Figure 8. Metabolic indices estimated between gut microbiome species and *Enterobacteriaceae*. **a**, Metabolic competition and complementary indices estimated with PhyloMint between co-excluders or co-colonizers and all *Enterobacteriaceae* species detected at >1% prevalence. **b**, Distribution of metabolic distance scores between co-colonizers ($n=4292$ comparisons) and co-excluders ($n=4773$ comparisons) in relation to *Enterobacteriaceae*. **c**, Comparison of metabolic distances within and between co-excluders and co-colonizers. Co-excluders vs. co-excluders: $n = 8256$ comparisons; co-colonizers vs. co-colonizers: $n = 6670$ comparisons; co-colonizers vs. co-excluders: $n = 14,964$ comparisons. **d**, Reproducibility of metabolic distance scores compared to *Enterobacteriaceae* after simulating models with defined gut media (M1) supplemented with diets from the Virtual Metabolic Human database, or with the M3 rich growth media. All comparisons were statistically significant ($P < 0.0001$). Box lengths represent the IQR of the data, with whiskers depicting the lowest

and highest values within 1.5 times the IQR of the first and third quartiles, respectively. *P* values were derived from a Wilcoxon rank-sum test.

4) Many of these associations make sense, and some are unsurprising. I suggest you focus a bit more in the discussion on which observations are truly novel/surprising.

Thank you for the suggestion. We have revised the discussion to focus more on the novel insights provided by our study, as follows:

Lines 386-397. “In addition to confirming previous associations (e.g., negative signal of *Enterobacteriaceae* with SCFA-producers) in a diverse large-scale cohort, our study provides several novel insights. First, we reveal significant and consistent microbiome differences associated with *Enterobacteriaceae* colonization across health states and geographic regions, uncovering a large uncharacterized subspecies diversity of *E. coli* among healthy adults in Africa. Moreover, our findings suggest that species from the *Faecalibacterium* genus beyond the well-established *F. prausnitzii* may play a critical role in colonization resistance against *Enterobacteriaceae*. We also identified notable co-colonization patterns involving underexplored taxa such as *Intestinibacter* and *F. phoceensis* that may underly novel biological mechanisms linked to *Enterobacteriaceae* outgrowth. Finally, we discovered that co-excluder species harbour a range of uncharacterized BGCs involved in quorum sensing that could be modulating *Enterobacteriaceae* abundance.”

5) Finally, I don't expect you to add experimental validation to this study (beyond the scope), but it would be nice to try to leverage existing data to validate some of your observations. For example, there are several fecal transplant studies that have stool metagenomic data from donors and recipients before and after treatment. Can the taxa/genes you identify here (or the ML models) predict which recipients will see engraftment of donor *Enterobacteriaceae* species/strains following treatment (and those who may exclude donor *Enterobacteriaceae* species/strains)?

We thank the reviewer for this great suggestion and we agree that comparing our data with existing studies would be a relevant way to evaluate some of our observations. To address this point, we first compiled various faecal microbiota transplant (FMT) datasets representing 1,341 metagenomic samples from 21 studies that were nicely summarized in Schmidt et al. *Nat. Medicine* 2022. We processed these samples through the same metagenomic analysis pipeline we employed for our dataset to quantify gut microbiome species abundance and prevalence. Thereafter, as suggested by the reviewer, we screened for FMT pairs where the donor was colonized by *Enterobacteriaceae*, but the recipient before FMT was not. Unfortunately, this filtering only revealed 4 eligible FMT pairs (2 that remained without *Enterobacteriaceae* after the FMT, and another 2 that acquired *Enterobacteriaceae* after the transplant). Given these low numbers it was not possible to perform further statistical analyses. We speculate that the reason why this FMT pairing is rare is because recipients tend to be individuals with a disrupted microbiome (e.g., with *C. difficile* diarrhoea or with inflammatory bowel disease), whereas donors are likely to be screened for the presence of multidrug-resistant pathogens, which include *Enterobacteriaceae*.

Nevertheless, as an alternative approach we decided to compare our results with another study (Kang et al. *Nat. Microbiol.* 2022, PMID: 36109646) not previously included in our dataset, which investigated the longitudinal dynamics of carbapenemase-producing *Enterobacteriaceae* (CPE). This study tracked for up to 12 timepoints a cohort of CPE-positive

individuals and CPE-negative household controls ($n = 46$ subjects; 361 samples). We believe this dataset represents a very pertinent comparison point, as the close proximity of the household controls to CPE-positive individuals suggests their lack of CPE colonization is more likely attributable to colonization resistance mechanisms within the gut microbiome.

We processed the samples through the same pipeline we employed in our study and performed two independent comparisons: (a) CPE-positive samples compared to CPE-negative household controls; and (b) CPE-positive samples compared to CPE-negative samples from subjects that were recently CPE-positive. Interestingly, with comparison (a) there was a statistically significant overlap (χ^2 test, $P = 0.0071$) between the differentially abundant species detected in this study and those found in our work. However, the overlap was more modest for comparison (b), which as discussed in the original CPE study, is likely due to the fact that individuals recently colonized by CPE are still in the process of recovering and transitioning towards a baseline healthy microbiome. We therefore suggest that the species overlapping between both studies are not only involved in *Enterobacteriaceae* colonization as a whole but may also be related with colonization of CPE strains in particular.

We have added these new results and a new figure (Extended Data Fig. 5, included below) in the revised manuscript, as follows:

Lines 214-227: “To further evaluate the clinical relevance of our findings, we compared our results with an independent study which investigated the longitudinal dynamics of carbapenemase-producing *Enterobacteriaceae* (CPE)¹⁰. This study tracked for up to 12 timepoints a cohort of CPE-positive individuals that were later decolonized, as well as CPE-negative household controls ($n = 46$ subjects; 361 samples). Comparison of differential abundant species between CPE-positive individuals and household controls showed a statistically significant overlap (χ^2 test, $P = 0.0071$) in relation to those we detected to be associated with *Enterobacteriaceae* in general (Extended Data Fig. 5a). However, the overlap was not significant when comparing CPE-positive to CPE-negative individuals that were previously colonized. This aligns with findings from the original CPE study¹⁰, which noted that individuals recently colonized by CPE are still undergoing microbiome recovery. Nevertheless, we suggest that the overlapping co-excluder and co-colonizer species here discovered (Extended Data Fig. 5b) are not only involved in *Enterobacteriaceae* colonization as a whole but may also be related with colonization of CPE lineages in particular.”

Lines 750-763: “Results were further compared with the dataset of Kang et al.¹⁰ that specifically investigated the longitudinal dynamics of carbapenemase-producing *Enterobacteriaceae* (CPE). A total of 361 samples were processed from study ERP133829 and mapped to the UHGG as described above. To identify differentially abundant species while accounting for the longitudinal study design, we used the mixed effects modelling implemented in MaAsLin2. The CPE status (CPE-positive, CPE-negative control or CPE-negative index) was used as a fixed effect, and the individual subject was used as a random effect. Samples from individuals who received antibiotics or were hospitalized since their last visit were excluded from the analysis. As above, the minimum prevalence threshold was set at 1% and statistical significance was determined with an FDR < 5%. The overlap with the species identified as *Enterobacteriaceae* co-excluders or co-colonizers was inferred with a χ^2 test using four estimates: i) number of species associated with CPE and *Enterobacteriaceae*; ii) number of species associated only with CPE; iii) number of species associated with *Enterobacteriaceae* but not CPE; iv) number of species not associated with either.”

Extended Data Figure 5. Co-excluders and co-colonizers of carbapenemase-producing *Enterobacteriaceae*. **a**, Number of species differentially abundant between individuals colonized by carbapenemase-producing *Enterobacteriaceae* (CPE) compared to household negative controls (left) and compared to CPE-negative index subjects that were decolonized within the previous year (right). Species are coloured based on whether they were also found to be significantly different, and in the same direction, using the whole *Enterobacteriaceae* family (green), missing (grey) or significant but in opposite directions (red). **b**, Effect size of species that were associated with both *Enterobacteriaceae* and CPE status using household controls (left) or using CPE-negative index subjects (right). Positive effect size denotes co-colonizers, while co-excluders are shown with a negative effect size. Error bars represent the standard error.

Reviewer #3 (Remarks on code availability):

While I have not cloned the github repo and tried to rerun the code, I looked at the repository and it appears to be reasonably well-documented and thorough. The link is functional, and it appears all the analysis code is accounted for.

We thank the reviewer for checking this. We have also added additional scripts based on the new analyses performed in the revised manuscript.